# Vivianite formation in methane-rich deep-sea sediments from the South China Sea

Jiarui Liu[1], Gareth Izon[2], Jiasheng Wang[1], Gilad Antler[3,4,5], Zhou Wang[1], Jie Zhao[1], Matthias Egger[6]

[1] State Key Laboratory of Biogeology and Environment Geology, College of Marine Science and Technology, School of Earth Sciences, China University of Geosciences, Wuhan, 430074, China
[2] Department of Earth, Atmospheric and Planetary Sciences, Massachusetts Institute of Technology, Cambridge, MA, 02139, USA
[3] Department of Earth Sciences, University of Cambridge, Cambridge, CB2 3EQ, UK
[4] Department of Geological and Environmental Sciences, Ben-Gurion University of the Negev, Beersheba, 84105, Israel
[5] The Interuniversity Institute for Marine Sciences, Eilat, 88103, Israel
[6] The Ocean Cleanup Foundation, Rotterdam, 3014 JH, the Netherlands

*Correspondence to*: Jiasheng Wang (js-wang@cug.edu.cn)

**Abstract.** Phosphorus is often invoked as the ultimate limiting nutrient, modulating primary productivity on geological timescales. Consequently, along with nitrogen, phosphorus bioavailability exerts a fundamental control on organic carbon production, linking all the biogeochemical cycles across the Earth system. Unlike nitrogen that can be microbially fixed from an essentially infinite atmospheric reservoir, phosphorus availability is dictated by the interplay between its sources and sinks. While authigenic apatite formation has received considerable attention as the dominant sedimentary phosphorus sink, the quantitative importance of reduced iron-phosphate minerals, such as vivianite, has only recently been acknowledged and their importance remains under-explored. Combining microscopic and spectroscopic analyses of handpicked mineral aggregates with sediment geochemical profiles we characterize the distribution and mineralogy of iron-phosphate minerals present in methane-rich sediments recovered from the northern South China Sea. Here, we demonstrate that vivianite authigenesis is pervasive in the iron oxide-rich sediments below the sulfate-methane transition zone (SMTZ). We hypothesize that the downward migration of the SMTZ concentrated vivianite formation below the current SMTZ. Our observations support recent findings from non-steady state post-glacial sedimentary successions, suggesting that iron reduction below the SMTZ, probably driven by iron-mediated anaerobic oxidation of methane (Fe-AOM), is coupled to phosphorus cycling on a much greater spatial scale than previously assumed. Calculations reveal that vivianite acts as an important burial phase for both iron and phosphorus below the SMTZ, sequestering approximately half of the total reactive iron pool. By extension, sedimentary vivianite formation could serve as a mineralogical marker of Fe-AOM, signalling low-sulfate availability against methanogenic and ferruginous backdrop. Given that similar conditions were likely present throughout vast swathes of Earth history, it is possible that Fe-AOM and vivianite authigenesis may have modulated methane and phosphorus availability on the early Earth, as well as during later periods of expanded marine oxygen deficiency. A better understanding of vivianite authigenesis, therefore, is fundamental to test long-standing hypotheses linking climate, atmospheric chemistry and the evolution of the biosphere.

# 1 Introduction

Phosphorus (P) is an essential nutrient, and its availability limits primary production on both short and long timescales (Algeo and Ingall, 2007; Ruttenberg, 2014). Marine sediments are known to regulate water column P availability, either retaining or releasing P dependent on the prevailing redox conditions (Delaney, 1998; Slomp et al., 1996). Phosphate is predominantly found as $HPO_4^{2-}$ in seawater (hereafter termed $PO_4$), which is captured and shuttled to the seabed in association with organic debris (organic P) or adsorbed onto iron (Fe)-(oxyhydr)oxides (hereafter termed Fe-oxide bound P). Through a combination of organic matter remineralization and reductive dissolution of Fe-oxides, $PO_4$ is released into the sediment pore water where precipitation of P-bearing minerals has the potential to sequester P over potentially geologically relevant timescales (Jensen et al., 1995; Ruttenberg and Berner, 1993; Sundby et al., 1992). Authigenic carbonate fluorapatite (CFA) is typically assumed to be the dominant sedimentary P mineral, accounting for around half of global marine P burial (Ruttenberg, 2014). Another potentially important group of P burial phases, that have only recently been recognized, are Fe(II)-phosphate minerals such as vivianite ($Fe_3(PO_4)_2 \cdot 8H_2O$). The global importance of these phosphates, however, remains under-constrained.

Vivianite authigenesis requires pore water with elevated ferrous iron ($Fe^{2+}$) and $PO_4$ concentrations. Low-sulfate lacustrine settings typically satisfy these criteria, and vivianite is commonly reported from freshwater sediments (e.g., Fagel et al., 2005; Rothe et al., 2014; Sapota et al., 2006). In more sulfate-rich settings, the presence of a sulfate-methane transition zone (SMTZ) has been shown to provide favorable conditions for vivianite authigenesis (Egger et al., 2015a, 2016; Hsu et al., 2014; März et al., 2008a, 2018; Slomp et al., 2013). The production of dissolved sulfide by sulfate-dependent anaerobic oxidation of methane ($SO_4$-AOM) in the SMTZ, and the associated conversion of Fe-oxides to Fe-sulfides, results in elevated pore water $PO_4$ concentrations around the SMTZ (März et al., 2008a). The subsequent downward diffusion of $PO_4$ into sulfide-depleted pore water below the SMTZ can then lead to the precipitation of vivianite, if sufficient reduced Fe is available at depth (e.g., Egger et al., 2015a; März et al., 2018).

Possible sources of $Fe^{2+}$ below the SMTZ are organoclastic Fe reduction, abiotic reductive dissolution of ferric-phases by sulfide, anaerobic oxidation of methane coupled to the reduction of ferric Fe (Fe-AOM), as well as more cryptic and less well understood mechanisms (Egger et al., 2017 and references therein). Of these, Fe-AOM is often advocated as the most likely mechanism, supplying plentiful $Fe^{2+}$ at the expense of methane due to the 8:1 stoichiometric conversion of Fe to methane (Eq. 1; Amos et al., 2012; Beal et al., 2009; Crowe et al., 2011; Egger et al., 2015b, 2016, 2017; Norði et al., 2013; Riedinger et al., 2014; Segarra et al., 2013; Sivan et al., 2011; Wankel et al., 2012).

$$CH_4 + 8Fe(OH)_3 + 15H^+ \rightarrow HCO_3^- + 8Fe^{2+} + 21H_2O \quad (Eq.1)$$

Iron-mediated AOM can be performed by anaerobic methane oxidizing archaea (ANME) who oxidize methane nonsyntropically, exploiting soluble and nanophase ferric iron ($Fe^{3+}$) as electron acceptors (Ettwig et al., 2016; Scheller et al., 2016). The presence of large multi-haem cytochromes (proteins that mediate electron transport) and type IV pili (cellular appendages) detected in the genomes of ANME hint that these archaea may also be able to exploit solid Fe-oxides via extracellular electron transport (McGlynn et al., 2015; Wegener et al., 2015). Moreover, the experimental approach of Bar-Or

et al. (2017) reveals that more refractory reactive Fe-oxides (e.g., magnetite and hematite) can also serve as electron acceptors for Fe-AOM. While biochemical investigations of *Methanoperedens ferrireducens* and *Methanosarcina activorans* have further illuminated the possible mechanisms of Fe-AOM (Cai et al., 2018; Yan et al., 2018), the modes and pathways of Fe-AOM remain enigmatic and warrant further exploration.

Vivianite authigenesis, potentially fueled by Fe-AOM, couples the biogeochemical cycles of Fe, P, sulfur (S) and carbon (C) in the deep biosphere. Given the importance of these elemental cycles within the Earth System, knowledge about vivianite precipitation and preservation in marine sediments is essential. While the database of vivianite occurrences in marine systems is growing, most of these records are from sites with atypical and time-variable stratigraphic records. For example, much of the work to-date has focused on vivianite authigenesis in non-steady state post-glacial sedimentary successions from marginal

basins like the Baltic and Black Seas (Dijkstra et al., 2016, 2018a, 2018b; Egger et al., 2015a; Reed et al., 2016). In these settings, post-glacial sea-level rise resulted in the accumulation of organic rich sediments overlying organic-poor lacustrine deposits. Other reports of possible vivianite formation in marine systems are from deep-sea fan sediments (Burns, 1997; März et al., 2008a, 2018), as well as from accretionary wedge sediments (Hsu et al., 2014). Consequently, a more detailed understanding of vivianite authigenesis, and the interplay between $CH_4$, S, Fe and P in open marine sediments, will improve

our ability to read ancient sedimentary records and to more adequately test hypotheses linking nutrient availability, climate and biospheric evolution.

    In this study, we combine microscopic and spectroscopic analyses of handpicked mineral aggregates with bulk geochemical analyses to characterize the distribution and mineralogy of Fe-phosphate minerals in the methane-rich deep-sea sediments preserved in the Taixinan Basin, northern South China Sea (Fig. 1). X-ray diffraction and down core abundance records reveal

vivianite authigenesis in this open marine sedimentary system. We further discuss the pathways for vivianite authigenesis below the SMTZ and the role of anaerobic methane oxidation. Our results support recent findings that vivianite formation can be an important burial mechanism for Fe and P in methane-rich marine sediments, indicating that vivianite authigenesis may have been more pervasive in the Earth's past.

## 2 Geological background and study site

The Taixinan (or Southwestern Taiwan) Basin is located east of the northern continental slope of the South China Sea (Fig. 1), separating a passive margin and an active accretionary wedge (Liu et al., 1997). As a Cenozoic hydrocarbon-bearing sedimentary basin featuring 1–4 km of sediment accumulation, the Taixinan Basin represents a promising area for gas hydrate and cold seep exploration (McDonnell et al., 2000). For example, a large seep-induced carbonate buildup (Jiulong Methane Reef, Site 1–3), covering about 430 km², was discovered during the R/V *SONNE* Cruise SO-177 in 2004 (Han et al., 2008;

Suess et al., 2005). East of these inactive seeps, Site F on the Formosa Ridge represents one of the most vigorous cold seeps reported from within the South China Sea, supporting a large and diverse chemosynthetic ecosystem (Feng and Chen, 2015; Feng et al., 2015, 2018; Hsu et al., 2017). Moreover, massive gas hydrates were documented during China's second major gas

hydrate expedition (GMGS-2) in 2013 (Sha et al., 2015; Zhang et al., 2015b), which further confirms that gas hydrates are well developed in the Taixinan Basin. Accordingly, given that the position of the SMTZ is governed by the flux of methane-rich fluids from depth, such a methane-rich area provides an ideal natural laboratory to explore how millennial-scale variations in methane flux influence vivianite distribution in continental margin settings.

This contribution exploits a piston core (total length of 13.85 m) that was taken at Site 973-4 (118°49' E, 21°54' N) in 2011 during a cruise with R/V *Ocean VI* (Fig. 1). Retrieved from the lower continental slope at a water depth of 1666 m, core 973-4 is dominated by dark-green silty clay. The only deviation in grain-size is a siltier layer between 455−605 cm depth, whose coarse fraction (> 65 µm) increases in association with increased abundances of foraminifera and Fe-rich silicates (Fig. 2a). Radiocarbon- and $\delta^{18}$O-derived age models (Fig. 2a) suggest sedimentation rates were relatively constant (32 cm ka$^{-1}$) throughout the ~40-thousand-years (ka) of deposition encompassed by core 973-4. Again, the only departure is associated with the coarser layer deposited during the Last Glacial Maximum, which yields atypically old radiocarbon ages relative to the surrounding sediment. These observations are consistent with increased slumping and/or turbidity currents (e.g., Zhong et al., 2015) driven by gas hydrate destabilization on the upper continental slope, and concomitant continental-slope failure during sea-level low stands (Kennett et al., 2003; Maslin et al., 2004).

**3. Methods**

After retrieval, core 973-4 was cut into sections and stored below 4 °C. These core sections were then split, subsampled and frozen (−20 °C) immediately after the cruise. Frozen samples were divided into two subsamples: the first subsample was used to isolate specific mineral phases via conventional handpicking (e.g., Lin et al., 2016a) and the second was homogenized for chemical analyses. All sample powders were kept frozen to minimize oxidation.

After drying at 60 °C for 24 h, the first set of subsamples were sieved with distilled water allowing the coarse fraction (> 65 µm) to be collected. Mineral aggregates were then identified and handpicked from the greater than 65 µm fraction (coarse component) under a stereomicroscope. The weight of these mineral fractions, along with the total coarse component, were determined and their concentrations were expressed relative to the initial dry mass of the sample. The morphology and chemical composition of the handpicked minerals were investigated using a FEI Quanta 450 FEG scanning electron microscope (SEM) in energy dispersive spectroscopy (EDS) mode. Power X-ray diffraction (XRD) analysis of the handpicked samples was performed using Ni-filtered Cu Kα radiation on a Panalytical X'Pert Pro diffractometer. The X-ray diffractometer was operated at 40 kV and 40 mA over a 3–65° 2θ range. The analytical step size was 0.017° with a measurement dwell time of 0.4 s per step. Raman analysis of the handpicked samples was performed by a JY/Horiba LabRam HR Raman system, using 532.06 nm (frequency doubled Nd:YAG) laser excitation, a 50× Olympus objective, and a 300-groove/mm grating.

Two different operationally defined solid-phase Fe pools were determined chemically exploiting separate aliquots of freeze-dried sample (e.g., Holmkvist et al., 2011, 2014). Briefly, the most readily acid-soluble Fe phases, including amorphous Fe (hydro)oxides and ferrihydrite (Haese et al., 1997; Wallmann et al., 1993), were extracted via agitation with an anoxic 0.5 M

HCl solution for one hour. The Fe(II) content of this extract was then determined via the 1, 10-phenanthroline method (Amonette and Templeton, 1998), followed by the total Fe content (i.e., Fe(II) + Fe(III)) with a 1, 10-phenanthroline and 1% (w/v) hydroxylamine hydrochloride assay. The poorly crystalline Fe(III) (oxy)hydroxide content was then calculated as the difference between the reduced and mixed-valence Fe determinations. More crystalline Fe-oxides, including goethite, hematite and part of the poorly reactive sheet silicate Fe fraction, were then quantified after treating a separate sample aliquot with a mixed dithionite-citrate-acetic acid solution for two hours (Poulton and Canfield, 2005). The Fe(III) and manganese (Mn) contents of the resulting supernatants were measured by atomic absorption spectrometry (Pgeneral, TAS-990). Replicate analysis of samples and standards (CUG-2 and CUG-3, Zhang et al., 2018d) displayed relative standard deviations (RSD) of better than 5%. Compared to the more well-documented dithionite method, the HCl method has not been as well calibrated. Therefore, for the ensuing discussion, we refer to dithionite-extractable Fe and Mn as reactive Fe-oxides and reactive Mn, respectively.

Pore water samples from Site 973-4 were not immediately extracted; instead, they were obtained several months later via centrifugation of previously frozen subsamples (Zhang et al., 2014). Consequently, we believe that the pore water chemistry of Site 973-4 may have been compromised, and is more likely to reflect post-recovery oxidation and/or contamination rather than in-situ sediment processes (Fig. 2b). Fortunately, pore water sulfate and methane concentrations are better constrained at several surrounding sites (Site DH-CL11, Site B and Site HD319), which are separated from Site 973-4 by less than a few kilometers (Lin et al., 2017b; Lu et al., 2012; Ye et al., 2016). Like Site 973-4, the adjacent sites are lithologically similar and dominated by silty clay. Moreover, the pore water chemistry from the surrounding sites is broadly analogous (Fig. 2b), indicating that each site was likely subject to the same depositional process(es). Consequently, in the absence of robust pore water data, we synthesize observations from these neighboring sites to estimate the position of the SMTZ at Site 973-4. We stress that we do not advocate this approach to replace pore water analysis but, in its absence, we argue that pore water data from nearby sites, when combined with solid-phase distributions from Site 973-4, allow us to estimate the approximate position of the SMTZ, albeit with caveats. Pore water extraction from Site DH-CL11 was conducted on shore via centrifugation, exploiting sample aliquots that had been immediately taken and stored under vacuum at −80 °C (Lin et al., 2017). Pore water samples from Site B and Site HD319 were collected immediately after recovery by vacuum extraction (Lu et al., 2012; Ye et al., 2016). At each site, the pore water sulfate contents were determined by ion chromatography; whereas, pore water methane concentrations were determined via sediment plug sampling and gas chromatographic analysis of the resultant headspace gas (Lin et al., 2017; Lu et al., 2012; Ye et al., 2016). The pore water chemistry of these adjoining sites reveals consistently shallow SMTZs, found between ~700 and 880 cm depth, which, via extension, is where we tentatively place the present-day SMTZ at Site 973-4.

Previously published data was determined using separate aliquots of the same subsamples exploited herein. This earlier work followed established protocols and full methodological details are provided in the respective papers. Accordingly, only a brief description is provided here: Acid volatile sulfide was liberated via HCl distillation and trapped by zinc acetate, its concentration was then determined spectrophotometrically (Zhang et al., 2014). Pyrite aggregates were handpicked from, and

expressed relative to, the coarse-fraction (> 65 µm). The sulfur isotopic composition of handpicked pyrite was determined directly via flash combustion using a Delta V Plus isotope ratio mass spectrometer (IRMS) interfaced with a Flash elemental analyzer (Lin et al., 2015). The solid-phase distribution of P was revealed through the SEDEX sequential extraction scheme (Ruttenberg, 1992). Iron-bound P and authigenic carbonate fluorapatite were extracted by citrate-bicarbonate-dithionite and

Na-acetate buffer, respectively (Zhang et al., 2018b). The carbon isotopic composition of total inorganic carbon was determined using a Finnigan MAT-252 IRMS after initial treatment with phosphoric acid (Ou, 2013; Zhang et al., 2018c). Finally, magnetic susceptibility data was generated using a MFK1-FA kappameter (Lin et al., 2017a).

## 4. Results

### 4.1 Morphology and chemical composition of mineral aggregates

Two types of dark mineral aggregates were identified. Examination of these handpicked mineral aggregates under a stereomicroscope revealed that the opaque blue to black crystals (0.1–1 µm diameter) were only present in samples below ~920 cm sediment depth (Fig. 3a). By contrast, stereomicroscope and SEM observations showed that pyrite is pervasive throughout the upper part of the core, observed in samples at ~310–880 cm depth. Moreover, SEM observations of the blue to black mineral phase found below the SMTZ disclosed a distinctive morphology, displaying spherical aggregates of radiating

lath-, platy- and needle-shaped crystals (Figs. 3b−e, g). This distinct morphology strongly resembles vivianite crystals identified from the Baltic Sea (Dijkstra et al., 2016; Egger et al., 2015a) and lacustrine sediments (Rothe et al., 2014, 2015). Major peaks of O, Fe, P were observed in the EDS spectra of the blue to black aggregates (Fig. 3f), constituting 39%, 28%, and 19% of the aggregates by mass (n=12), respectively. The O, Fe, P mass-ratios and the molar Fe/P ratio (0.83) from EDS analyses also approximate those reported from sedimentary vivianite from the Bothnian Sea (0.82 and 0.86), as well as

synthesized vivianite (0.99; Egger et al., 2015a). Additional minor peaks of Mg, Si, Al, Ca, Mn, S were also observed in the EDS spectra, constituting 4.9%, 1.8%, 1.6%, 1.6%, 1.2%, 1.2% of the aggregates by mass (n=12), respectively. The Fe–Mg– Mn ternary plot of these aggregates (Fig. 4) reveals that the Fe/(Fe+Mg) ratios range from ~0.64 to 0.98, while the Mn/(Mn+Fe) ratios range from ~0 to 0.05. These observations are similar to the high-Mg, low-Mn vivianite identified in the sediments recovered from offshore southwestern Taiwan (Hsu et al., 2014).

Additional grey to green mineral aggregates were observed in samples throughout the sediment core. These particles feature smooth surfaces (Fig. 3h), while their EDS spectra show high intensity Fe, Si, O peaks (Fig. 3i), suggesting an Fe-rich silicate phase.

### 4.2 XRD and Raman analyses of mineral aggregates

X-ray diffraction analyses confirm that the blue to black mineral aggregates found below the SMTZ are vivianite nodules (Fig.

5a). The high intensity narrow peak observed at 13.2° typifies XRD spectra obtained from both natural and synthetic vivianite crystals (Dijkstra et al., 2016; Egger et al., 2015a; Grizelj et al., 2017; Rothe et al., 2014). Other Mg- or Fe-bearing phosphate

minerals such as metavivianite were not recognized in diffractograms of our samples. Raman analyses showed a curved baseline with broad peaks (Fig. 5c), potentially caused by fluorescence interference from the samples (Kagan and McCreery, 1994). Despite these limitations, some peaks can still be recognized, especially the peak around 1000 cm[-1]. These spectral features are also consistent with the Raman spectra of vivianite crystals (Piriou and Poullen, 1984), and thus offer an additional line of evidence supporting the results obtained from SEM-EDS and XRD analyses.

The grey to green mineral aggregates are mainly composed of chlorite, illite, quartz, and albite. Additional peaks including orthoclase, siderite, calcite, dolomite, pyrite and vivianite were also identified (Figs. 5b and 6). The XRD spectra indicated that some illite peaks might belong to mixed-layer illite/smectite. Glauconite is a green Fe-rich member of illite group and is an abundant authigenic mineral at water depths between 30 and 2000 m (Porrenga, 1967). Considering its EDS spectra, with high intensity Fe, Si, O peaks, it is likely that the illite-group minerals identified in core 973-4 may include a considerable glauconite component. Besides Fe-rich illite, or possibly even glauconite, chlorite is another Fe-rich sheet silicate that would also give similar EDS spectra. Given the difficulties in distinguishing between low abundance Fe-rich clays in complex marine sediments by XRD, the grey to green mineral aggregates are simply referred to as Fe-rich silicates from here on.

### 4.3 Down-core variations of mineral aggregates

The distribution of authigenic minerals at Site 973-4 displays distinct down-core variability, especially at and around the SMTZ. Substantial coarse grained pyrite was observed between ~560–880 cm depth with a peak around the SMTZ. Handpicked vivianite aggregates were only identified between ~920–1370 cm depth in the sediment (Fig. 7a). The concentration of vivianite ranges from 0.02 wt.‰ to 1.58 wt.‰ directly below the SMTZ (920–1175 cm depth), decreasing to lower values (0 to 0.36 wt.‰) towards the base of the core (1175–1370 cm depth). Dark laminations and reddish-brown nodules are prominent between 892–904 cm depth where acid volatile sulfur (AVS) displays a sharp peak at the base of the SMTZ (Fig. 9b). Major peaks of O and Fe were observed in the EDS spectra of these nodules, while lower intensity S peaks were only observed in a few samples. Nodules of AVS usually turn reddish-brown during post-sampling oxidation, thus most of the nodules found at the base of the SMTZ are most likely oxidation products of AVS.

Fe-rich silicates were distributed throughout the core (Fig. 7b), displaying a peak between 455–605 cm depth. This layer has an elevated coarse-grained component and is associated with atypically old radiocarbon ages (Fig. 2a). The XRD spectra reveal that the Fe-rich silicates within this layer include more chlorite, differing from the composition of background Fe-rich silicates (Fig. 6). Compared with the high abundances of Fe-rich silicates observed above (8.48 wt.%) and below (2.94 wt.%) the SMTZ, the abundance of Fe-rich silicates within the SMTZ is servilely diminished, ranging from 0.01 wt.% to 0.16 wt.%.

### 4.4 Solid phase iron and manganese geochemistry

Ferric iron minerals were quantified using two different extraction methods, with both approaches yielding broadly consistent trends (Fig. 8a). The absolute concentrations of Fe (oxy)hydroxides and reactive Fe-oxides are low immediately above and within the SMTZ, reaching a minimum of 34 µmol g[-1] and 76 µmol g[-1] within the more silty layer, respectively. Their respective

concentrations increase to 173 µmol g$^{-1}$ and 238 µmol g$^{-1}$ at 900 cm depth and remain high around 123–161 µmol g$^{-1}$ and 174–235 µmol g$^{-1}$ below the SMTZ (Fig. 8a). Since more crystalline Fe minerals (e.g. goethite, hematite) are extracted by the dithionite-citrate-acetic acid solution (Poulton and Canfield, 2005; Raiswell et al., 1994), the concentration of reactive Fe-oxides is higher than that of Fe (oxy)hydroxides throughout the core. Therefore, for simplicity, reactive Fe-oxides will be referred to as Fe-oxides in the following discussion. Reactive Mn concentrations range from 1.6 µmol g$^{-1}$ to 4.6 µmol g$^{-1}$, which is nearly two orders of magnitude higher than those of Fe (oxy)hydroxides and reactive Fe-oxides. Fe and Mn show similar distribution patterns with depth (Fig. 8). Since AVS oxidizes to dithionite and oxalate extractable Fe phases during freeze-drying or exposure to air (Canfield, 1989; Morse, 1994), the obvious peak of Fe-oxides at ~900 cm depth (Fig. 8a) is partly attributed to AVS oxidation during sample storage and treatment.

## 5. Discussion

### 5.1 The role of the SMTZ in marine vivianite authigenesis

Vivianite precipitation is favored in anoxic and non-sulfidic settings; conditions that are frequently encountered in lacustrine sediments (Berner, 1981; Nriagu, 1972; Rothe et al., 2016). Recently, the presence of a SMTZ has been shown to play a principle role in vivianite formation, extending the importance of vivianite authigenesis to marine settings where sulfate is more readily available (Egger et al., 2015a; Hsu et al., 2014; März et al., 2008a; Slomp et al., 2013). Within the SMTZ, sulfate-driven AOM consumes sulfate and methane whilst liberating sulfide and bicarbonate. The resultant sulfide is rapidly fixed, initially as Fe monosulfides and ultimately, under an excess of sulfide, as pyrite, acting as a permanent sink for sulfur. Therefore, given a constant methane flux, the locus of Fe sulfide precipitation is governed by the depth to which sulfate penetrates the sediment pile. Pyrite formation within the SMTZ is characterized by a strong $^{34}$S enrichment along with characteristic overgrowth textures (Borowski et al., 2013; Jørgensen et al., 2004; Lin et al., 2016a; Lin et al., 2016b). In addition, sulfate driven AOM enriches authigenic carbonates in $^{12}$C, as they inherit their carbon isotopic signature from $^{13}$C-depleted methane (Peckmann and Thiel, 2004; Treude et al., 2005). At Site 973-4, between ~560 and 880 cm depth, the pyrites display a pronounced increase in $\delta^{34}$S values in concert with an excursion to low $\delta^{13}$C values registered in the total inorganic carbon pool (Fig. 9). The ingrowth of these distinctive isotopic signatures within the solid-phase record at Site 973-4 requires that the SMTZ has remained stable between 560–880 cm depth for millennia (März et al., 2008a). Caused by the conversion of Fe-oxides to pyrite, the drop in magnetic susceptibility at the same depths (Fig. 8b) provides support for this inference. In detail, however, precisely constraining the depth of the current SMTZ at Site 973-4 is more difficult, complicated by the lag-time of the solid-phase record and the absence of reliable pore water data (discussed below). Nevertheless, complications aside, the SMTZ depth-estimate derived from Site 973-4's solid-phase record is broadly consistent with that inferred from neighboring pore water profiles (Fig. 2b).

Reactive Fe-oxides and Fe-rich silicates are gradually consumed within the SMTZ via reductive dissolution by sulfide and subsequent conversion to pyrite (Figs. 7b and 8a). Consequently, any excess sulfide descending from the SMTZ is trapped as

Fe monosulfides upon reaction with more readily available Fe-oxides at depth. This reaction zone, often referred to as the sulfidization front (S-front), is a common feature in Fe-rich marine sediments (Egger et al., 2016; Holmkvist et al., 2014; Jørgensen et al., 2004; Riedinger et al., 2017) and is recorded at Site 973-4 as a pronounced AVS peak at ~900cm depth (Fig. 9b). Besides liberating Fe, and 'seeding' Fe-sulfides, reductive dissolution of Fe-oxides by sulfide also releases a significant amount of Fe-oxide bound P into the pore water around the SMTZ (Egger et al., 2015a; März et al., 2008a; Schulz et al., 1994; Slomp et al., 2013). Previous studies have demonstrated that $PO_4$ concentrations typically exceed ~100 µM in the SMTZ when vivianite is observed, providing empirical evidence for a SMTZ-derived P source for subsurface vivianite precipitation (e.g. Egger et al., 2015; März et al., 2008a, 2018). Unfortunately, there is no reliable $PO_4$ pore water profile from Site 973-4; however, the conversion of Fe-oxides to pyrite and the associated release of Fe-oxide bound P to the pore water should also be documented within the solid-phase records (Egger et al., 2015a, 2016). Consequently, the distinct minimum of Fe-bound P around the SMTZ, and a sharp peak directly below the SMTZ, strongly suggests that sink-switching from Fe-oxide bound P to reduced Fe-phosphates is occurring at Site 973-4 (Fig. 9b).

X-ray diffraction analyses show that the blue to black aggregates found exclusively below the inferred position of the SMTZ are vivianite aggregates (Fig. 5a). Other Mg- or Fe-bearing phosphate minerals such as metavivianite were absent from the XRD spectra, advocating that the Fe-P-O aggregates identified by SEM-EDS and Raman are indeed vivianite (Figs. 3 and 5). The restriction of vivianite nodules to below the SMTZ strongly supports the SMTZ-catalyzed model of vivianite authigenesis resulting from inferences gleaned from the juxtaposition of lacustrine and post-glacial marine sediments (Egger et al., 2015a, 2016; Slomp et al., 2013). Furthermore, the broad peak in Fe-bound P above the SMTZ likely reflects retention and re-adsorption of ascending $PO_4$ to shallower Fe-oxides (Fig. 9b), as suggested previously based on observations from the Zambezi deep-sea fan (März et al., 2008a).

Accepting the uncertainties and extending the pore water chemistries from adjacent sites to Site 973-4, suggests that the SMTZ currently resides around 700–880 cm depth (Fig. 2b), deeper than one would predict based on the solid-phase records alone. This discrepancy is most likely due to the different response times of the aqueous- and solid-phase records, implying that the SMTZ has not been stationary over the cored interval. We reconcile the preservation of Fe-oxides and the absence of pyrite below the SMTZ with a rapid ascent of the SMTZ. Such a rapid shallowing of the SMTZ would limit sulfide exposure, causing incomplete reduction and sulfidization of the deeper reactive Fe pool (März et al., 2018). Following its hypothetical ascent, the SMTZ must have then remained stable between 560–880 cm, leaving pronounced imprints within the pyrite, Fe-oxide, Fe-silicate, total inorganic carbon and magnetic susceptibility profiles (Figs. 7–9). Assimilating these observations, we suggest that the SMTZ descended from ~560–700 cm to its current position at ~700–880 cm depth. Given that the sedimentation rate at Site 973-4 was almost constant over the duration of the core (Fig. 2a), we attribute the hypothesized downward migration of the SMTZ to a change in the methane flux. Furthermore, we suggest that the evolution of the SMTZ reflects glacial–interglacial sea-level changes, with the sea-level low stand at Last Glacial Maximum decreasing hydrostatic pressures, destabilizing methane clathrates (Kvenvolden, 1993) and promoting a rapid upward migration of the SMTZ due to enhanced methane fluxes (Borowski et al., 1996). Conversely, the subsequent Holocene sea-level rise would have diminished methane

fluxes, instigating a slow downward migration of the SMTZ to its current position. These observations advocate that hydrate-derived methane fluxes are heterogeneous and, along with vivianite authigenesis, are likely to have been variable on millennial time-scales (Ruppel and Kessler, 2017; März et al., 2018).

Vivianite is reactive toward sulfide and, hence, would be readily converted to Fe-sulfide phases under $H_2S$-rich conditions (Berner, 1981). Therefore, when the SMTZ was shallower, we propose that vivianite was precipitated above where it is currently observed and has subsequently been dissolved. Tentative evidence for diagenetic modification of the original vivianite distribution is seen in our XRD data. Here, XRD analysis indicates the presence of vivianite within the current SMTZ at 747 cm depth, yet handpicking failed to identify any coarse vivianite (Figs. 6–7a). Dissolution of freshly precipitated vivianite should be promoted by the downward migration of the SMTZ and increased environmental sulfide availability, concentrating vivianite below the SMTZ and producing the observed stratigraphic distribution (Fig. 7a). Exactly why the vivianite at 747 cm depth survived sulfidization remains uncertain, however, the textual association with Fe-silicates implies that these phases may have armored the vivianite, preventing its conversion. In the event of the SMTZ shallowing, any sub-SMTZ vivianite could potentially be preserved providing the environment remained $H_2S$-free and no other mineral transformations occurred (see below). Thus, we hypothesize that long-term migration patterns of the SMTZ will alter the solid-phase record and control the distribution of vivianite. The integrative effect, of course, creates the mineralogical and geochemical distribution profiles we observe today.

## 5.2 The importance of anaerobic oxidation of methane in vivianite authigenesis

The ubiquity of vivianite aggregates below the SMTZ at Site 973-4 requires that there is a deep source of $Fe^{2+}$. Recent field and laboratory studies have suggested that anaerobic oxidation of methane can be coupled to the reduction of Fe-oxides (Beal et al., 2009; Egger et al., 2015b, 2016, 2017; Riedinger et al., 2014; Sivan et al., 2011). Specifically, it has been suggested that insoluble Fe-oxides can be exploited by solitary anaerobic methanotrophic archaea (ANME) as electron acceptors, facilitating methane oxidation via extracellular electron transfer (McGlynn et al., 2015; Rotaru and Thamdrup, 2016; Scheller et al., 2016; Wegener et al., 2015). Moreover, sediment incubation experiments demonstrate that more poorly reactive Fe minerals (e.g., magnetite and hematite) are also bioavailable and, therefore, could potentially fuel Fe-AOM (Bar-Or et al., 2017). Extending these in vitro observations to natural settings suggests that bioavailable Fe-oxides, and potentially reactive Fe-rich silicates below the SMTZ may be available for bacterial Fe reduction coupled to methane oxidation. Considering the 8:1 Fe-$CH_4$ stoichiometry (Eq. 1; Beal et al., 2009), Fe-AOM certainly has the potential to yield significant quantities of $Fe^{2+}$. In general, large amounts of dissolved $PO_4$ are released into the pore water by sulfate-driven AOM within the SMTZ (e.g., März et al., 2008a), while bacterial Fe reduction using Fe-oxides below the SMTZ triggers the release of both $Fe^{2+}$ and $PO_4$ (Egger et al., 2015a). Supported by abundant Fe-oxides and Fe-silicates below the SMTZ, bacterial Fe reduction, probably driven by methane oxidation, promotes vivianite authigenesis, exchanging P originally associated with Fe-oxides to vivianite-housed P (Fig. 10). As a byproduct, Fe-AOM is known to produce alkalinity (Beal et al., 2009), which can stabilize vivianite by raising the pH values of the pore water to between 6 and 9 (Rothe et al., 2016).

Besides Fe-AOM, other potential sources of $Fe^{2+}$ below the SMTZ include organoclastic Fe reduction (Lovley, 1997; Severmann et al., 2006) and abiotic reductive dissolution by sulfide (Canfield et al., 1992; Poulton et al., 2004). A pronounced AVS peak at ~900 cm depth constrains the depth of the sulfidization front at Site 973-4 (Fig. 9b). In similar systems, dissolved sulfide is limited below the S-front (Egger et al., 2016; Jørgensen et al., 2004; Riedinger et al., 2017). As vivianite is unstable in the presence of sulfide (Dijkstra et al., 2018a), its presence below ~920 cm depth supports the absence of appreciable amounts of dissolved sulfide. Additionally, extremely elevated concentrations of $Fe^{2+}$ are frequently observed below the S-front (Egger et al., 2016; Holmkvist et al., 2011, 2014; Jørgensen et al., 2004; Treude et al., 2014). Given the reactivity of $Fe^{2+}$ toward sulfide, the existence of the latter would rapidly titrate the former, forming Fe monosulfides rather than reacting with Fe-oxides (Berner, 1967). We therefore conclude that abiotic sulfide-mediated reductive dissolution of Fe-oxides is unlikely to provide significant amounts of $Fe^{2+}$ below the SMTZ.

Although unequivocally precluding organoclastic Fe reduction below the SMTZ remains difficult, the total organic carbon content throughout core 973-4 is low and more-or-less invariant, with an average value of $0.72 \pm 0.19$ wt.% (Zhang et al., 2014). These relatively low concentrations of likely reworked organic matter may be insufficient to fuel organoclastic Fe reduction (Riedinger et al., 2014). Consequently, the importance of organoclastic Fe reduction is likely to be limited by the quality and quantity of organic matter burial below the SMTZ (Egger et al., 2017; Riedinger et al., 2014; Sivan et al., 2011).

Besides its major constituents (Fe, O and P), sedimentary vivianite aggregates are usually enriched in other minor elements, such as Mg (Burns, 1997; Dijkstra et al., 2018b; Hsu et al., 2014) or Mn (Dijkstra et al., 2018b; Egger et al., 2015a; Fagel et al., 2005; Nakano, 1992; Sapota et al., 2006). The solubility product constant $K_{sp}$ of Mg phosphate is $10^{-24}$, while the $K_{sp}$ of vivianite is $10^{-36}$ (Nriagu, 1972), making the former mineral more soluble. Moreover, Mg concentrations are only about three orders of magnitude higher than those of Fe in marine sediments (e.g. Hu et al., 2015). Therefore, Mg phosphate precipitation is unlikely and $Mg^{2+}$ is more likely to be co-precipitated with vivianite below the SMTZ. In the absence of appropriate pore water data, we rely on datasets from a proximal site that reveals high concentrations of dissolved Mg (48.7–53.0 mM) but low concentrations of dissolved Mn (0.4–5.1 µM; Site D-5, Hu et al., 2015). These pore water chemistries are reflected in the elemental composition of the vivianite isolated from core 973-4 (Fig. 4). Manganese displays similar geochemical properties as Fe, yet it has been shown to be a substantially more energetically favorable electron acceptor during organic matter or methane oxidation (Beal et al., 2009). We reconcile the extremely low vivianite-housed Mn contents at Site 973-4 with the generally low sedimentary Mn abundances (Fig. 8b), concluding that Mn-mediated AOM was probably of limited importance Site 973-4. Taken together, while we cannot completely exclude organoclastic Fe reduction as a potential source of $Fe^{2+}$, along with observations of others, we use the discussed stratigraphic distribution of various sedimentary mineral phases to argue that Fe-AOM is the most likely deep source of $Fe^{2+}$ necessary to promote vivianite authigenesis observed below the SMTZ. Consequently, we speculate that where basal waters are oxygenated, sedimentary vivianite formation could serve as a mineralogical marker of Fe-AOM, signalling low-sulfate availability against methanogenic and ferruginous backdrop.

Important P and Fe phases that could compete with vivianite formation are authigenic apatite and Fe monosulfides. PHREEQC calculations of pore water saturation indexes (Egger et al., 2015a; März et al., 2018) show that hydroxyapatite saturation

indexes (SI) typically reach a maximum (above 0) around the SMTZ but drop to background values below the SMTZ, suggesting that the steady supply of $PO_4$ from reductive dissolution of Fe-oxides fosters thermodynamically favorable conditions for apatite authigenesis in proximity to the SMTZ. While there are no reliable pore water data from Site 973-4, dissolved $Ca^{2+}$ concentrations from Site B are consistent with ongoing apatite authigenesis, with a linear down-core decrease from the sediment-water interface to the upper part of the SMTZ (12.5–2.5 mM) where most of the $Ca^{2+}$ has been consumed (Ye et al., 2016). To our knowledge, however, there are no available dissolved $F^-$ and $PO_4$ data to unequivocally confirm apatite precipitation rather than other calcium-harboring phases. Higher SI for vivianite are generally obtained below the SMTZ, consistent with the vivianite distribution observed at Site 973-4. Importantly, vivianite authigenesis occurs below the sulfidization front (~900 cm) implying that FeS precipitation is kinetically favored over vivianite precipitation. We speculate, therefore, that the pore water chemistry and the activity of sulfidization front will influence the depth of vivianite formation. Intensified activity at the sulfidization front would liberate more $PO_4$. The concomitant sulfide flux, however, may serve to nullify the ascent of $Fe^{2+}$ from depth, especially as highly reactive Fe is consumed, preventing vivianite formation or, more likely, confining its authigenesis to deeper into the sediment pile. Unfortunately, we do not have the data to definitively test the hypothesized competing role between apatite, FeS and vivianite authigenesis. Nevertheless, any process that serves to deplete pore water $PO_4$ and $Fe^{2+}$ will hypothetically compete with vivianite formation. Future work should focus on generating comprehensive pore water and solid phase data sets, which when coupled with reactive transport models, will better link the rate of sulfide production in the SMTZ, the amount of reactive Fe-oxides and the generation potential of different authigenic P phases below the SMTZ.

**5.3 Quantifying the importance of vivianite as sedimentary sinks of iron and phosphorus**

Sequential P extractions (SEDEX; Ruttenberg, 1992), conducted by Zhang et al. (2018b), demonstrate that authigenic carbonate fluorapatite (Ca-P, Fig. 9c) accounts for up to 55% of the total P (702 µmol $g^{-1}$) buried at Site 973-4. Fe-bound P is generally lower than authigenic Ca–P, accounting for 27% and 13% of the total P burial above (613 µmol $g^{-1}$) and below the SMTZ (844 µmol $g^{-1}$), respectively. These results emphasize the role of Fe-bound P as an important mechanism for P burial in continental margin settings (Slomp et al., 1996). Interestingly, the ratio between Fe-oxides and Fe-bound P is also apparently depth dependent; decreasing from approximately 3 at the sediment surface, to 1.8 below the SMTZ. This decrease indicates a depth-dependent change in the relative contribution of authigenic phases responsible for P sequestration. Consistent with observational and geochemical evidence for the presence of authigenic vivianite (Fig. 7a), the decrease in the Fe-oxide/Fe-bound P ratio to 1.8 approximates the stoichiometric Fe/P of vivianite (1.5), again advocating that vivianite authigenesis below the SMTZ is a potentially important process responsible for P burial at Site 973-4.

Previous studies have shown that vivianite dissolves in both the citrate-dithionite-bicarbonate (CDB; Nembrini et al., 1983) and dithionite steps (Dijkstra et al., 2014) of the sequential P- and Fe-extractions, respectively. Thus, both fractions contain P and Fe derived from vivianite ($P_{viv}$ and $Fe_{viv}$) and Fe-oxides ($P_{FeOx}$ and $Fe_{FeOx}$). Consequently, the total amount of P extracted during the CDB step is the sum of $P_{viv}$ and $P_{FeOx}$, while the total amount of Fe extracted during the dithionite steps is the sum

of $Fe_{viv}$ and $Fe_{FeOx}$. Combining these measured fractions with the $Fe_{FeOx}/P_{FeOx}$ ratio of surface sediments at our site (~3) as a measure of P binding capacity and the stoichiometric Fe/P ratio of vivianite (1.5), we can estimate the quantitative importance of vivianite burial via the approach outlined by Egger et al. (2015a). Here, adopting average concentrations of Fe-bound P (108.3 µmol g$^{-1}$, Fig. 9b) and Fe-oxides (196.4 µmol g$^{-1}$, Fig. 8a) from below the sulfidization front at Site 973-4, a set of four equations, with an equal number of unknowns can be written:

$$Fe_{FeOx} + Fe_{viv} = 196.4 \text{ µmol g}^{-1} \quad \text{(Eq.2)}$$
$$P_{FeOx} + P_{viv} = 108.3 \text{ µmol g}^{-1} \quad \text{(Eq.3)}$$
$$Fe_{FeOx} = 3P_{FeOx} \quad \text{(Eq.4)}$$
$$Fe_{viv} = 1.5P_{viv} \quad \text{(Eq.5)}$$

Simultaneously solving these equations for $Fe_{viv}$ and $P_{viv}$ reveals that ~129 µmol g$^{-1}$ of the Fe extracted during the dithionite extraction, along with ~86 µmol g$^{-1}$ of the Fe-bound P, originated from vivianite, accounting for ~79% of the total Fe-bound P below the SMTZ. Authigenic Ca-P is the major P sink (~55%) at Site 973-4, in agreement with its globally estimated importance (~50%; Ruttenberg, 2014; Ruttenberg and Berner, 1993). Vivianite, by contrast, houses approximately 10% of the sub-SMTZ P inventory. These results further suggest that only 1 to 7.3% (average and maximum, n=70) of the chemically-constrained vivianite was recovered as coarse aggregates (Fig. 7a). This discrepancy, in turn, implies that much of the authigenic vivianite fraction is either disseminated as smaller crystals (< 65 µm) or as an amorphous solid-phase, as suggested previously by März et al. (2008a, 2018).

Vivianite plays an important role in Fe cycling below the SMTZ. The sediment column at Site 973-4 can be divided at ~900 cm depth, forming two distinct geochemical zones with different Fe, P and S systematics. Thus, several additional ratios can be calculated based on the previously discussed Fe- and P-extractions (Table 1). Total reactive Fe includes Fe-oxides and their authigenic reduction products (e.g., vivianite, pyrite and Fe carbonates). Considering that total Fe abundances (725 ± 30 µmol g$^{-1}$) are broadly invariant below 600 cm depth (Zhang et al., 2018b), and that the maximum AVS content (9.3 mmol g$^{-1}$) is one order of magnitude more abundant than total Fe, either we need to reconsider the quality of the AVS measurements (Zhang et al., 2014), or much of the quantified AVS (mainly $H_2S$ + FeS) reflects dissolved sulfide, rather than Fe sulfides (Fig. 9b). Given the difficulties in deconvolving the relative importance of FeS and dissolved sulfide, coupled with the likely restriction of FeS to the sulfidization front at ~900 cm depth (Fig. 9b), FeS is precluded from further discussion.

Moving forward, this approach reveals that 52% of Fe-oxides were likely reduced to pyrite within the SMTZ, whereas 41% were converted to vivianite below the SMTZ. While vivianite may represent a major component of total P inventory below the SMTZ in coastal settings (40–50%; Egger et al., 2015a), its importance as an Fe sink is muted, representing only a small fraction of the of total reactive Fe budget in the Baltic Sea sediments (~6.4%; Rooze et al., 2016). At Site 973-4, however, vivianite accounts for almost half of the reactive Fe burial below the SMTZ, arguing that vivianite may represent an important Fe burial phase in methane-rich deep-sea sediments. Considering the concentrations of total Fe are broadly constant below 600 cm depth at Site 973-4, Fe reduction contributes similar amounts of Fe(II) (~34% and ~32% of total Fe) below and within the SMTZ providing the Fe(II) phases are authigenic. Hence, besides abiotic reductive dissolution of Fe-oxides within the SMTZ,

Fe cycling is strongly affected by Fe reduction at depth. Based on the discussion above, we argue that Fe-AOM is a likely source of reduced Fe, which may sequester around 34% of total Fe as a reduced burial phase below the SMTZ. Accordingly, bacterial Fe reduction coupled to AOM likely plays an important role for Fe and P cycling at depth.

While the detailed mineralogical work necessary to confirm the spatial importance of vivianite authigenesis in open marine sediments awaits, at least at Site 973-4, vivianite authigenesis proceeds below the SMTZ, driven apparently by deep Fe reduction which, in the absence of free sulfide, apparently supplies copious quantities of pore water $Fe^{2+}$ and $PO_4$. Therefore, it follows that substantial burial of vivianite below the SMTZ associated with Fe reduction will have significantly altered the sedimentary record. Consequently, extra caution is warranted when using sedimentary P distributions of methane-rich sediments to reconstruct past ocean primary productivity and P burial pathways (Dijkstra et al., 2018a).

## 5.4 Long-term vivianite preservation—An outstanding question

The paleoenvironmental importance of vivianite in a geological sense depends on its longevity within the sediment pile. As discussed, vivianite is unstable in the presence of $H_2S$ yet the presence of an SMTZ serves as an efficient sulfide trap, and thus may promote vivianite preservation at depth. Consequently, there may be deeper layers of vivianite preserved within the 3–4 km thick sediment package preserved in the Taixinan Basin (McDonnell et al., 2000). Naturally, however, this assumption requires that vivianite does not undergo any further down-core transformation.

At Site 973-4 authigenic Ca-P concentrations increase down core (Fig. 9c), which may reflect the conversion of more labile P-species into authigenic apatite. This sink switching is considered to be the globally significant driver of apatite authigenesis in marine sediments (e.g. Ruttenberg and Berner, 1993; Slomp et al., 1996). Whether vivianite is converted to authigenic apatite at depth, while possible, remains an open question. If it were a simple conversion, however, one would predict a linear increase in authigenic apatite at the expense of vivianite. Closer inspection of the relevant solid phase records reveals that this is not the case and, in fact, the relationship between apatite and vivianite concentrations is ambiguous at Site 973-4 (Figs. 7a and 9c). Alternatively, the apparent increase in authigenic Ca-P from the SMTZ to the bottom of the core may reflect non-vivianite-related apatite authigenesis at depth. Upward fluxes of $Ca^{2+}$ and $F^-$ to the SMTZ have been observed in continental margin settings (e.g. Clemens et al., 2016; März et al., 2018), which would provide the necessary chemical constituents to promote sub-SMTZ apatite authigenesis. If apatite was the more favorable $PO_4$ sink, then precipitation of this mineral phase could hypothetically curtail vivianite authigenesis. In this scenario, although the two phosphate phases are linked, they are linked through $Ca^{2+}$- and $F^-$-availability rather than stability and conversion of one phase to another.

Besides potential transformation to authigenic apatite, vivianite is sensitive to oxidation and thus it is possible that vivianite in deep-time records has been transformed into Fe(III) phosphates (e.g. koninckite; März et al., 2008a), Fe-oxides (e.g. hematite; Berner, 1981), or some other unknown phase. Pseudomorphs of vivianite are apparently common in the deep-time record, providing at least some evidence for the transformation and stabilization of vivianite as it ages (e.g. metavivianite; Rodgers, 1986). Equally, however, the perceived absence of vivianite in aged sedimentary archives could be an artifact of insufficient surveys. Akin to März et al. (2018), we have shown that vivianite is a finely disseminated phase and thus may have escaped

detection in many studies. Moreover, typically employed extraction schemes co-extract vivianite, meaning that Fe-bound P in deep-time records could indeed have been wrongly ascribed, and could be vivianite (c.f., März et al., 2008b). Future work, targeting long cores, coupling aqueous- and solid-phase analysis, is required to definitively test our hypotheses linking vivianite authigenesis and Fe-AOM activity, as well as to examine the long-term stability of vivianite in the sedimentary record. More

routine application of XRD analyses and modification of solid-phase extraction protocols is also necessary to more completely understand the geological significance of vivianite.

## 5.5 Phosphorous cycling through Earth history

The redox state of a given water mass is expected to profoundly influence P-cycling. For example, where basal waters are oxygenated, a similar pore water pattern to that observed at Site 973-4, or other modern open marine setting, is generally

expected (Fig. 10), fostering vivianite authigenesis below the SMTZ. If oxygen concentrations drop, however, vivianite authigenesis is expected to be different with implications for P-cycling and productivity. Under ferruginous (anoxic and $Fe^{2+}$-rich) conditions, $PO_4$ would be adsorbed to and/or co-precipitated with Fe-oxides (März et al., 2008b), or even precipitated as vivianite directly if $Fe^{2+}$ and $PO_4$ concentrations were sufficiently high (Dijkstra et al., 2018b). This P shuttle is, at least, partially eradicated if euxinic conditions develop in the water column, caused by sulfidization of Fe-oxides and/or vivianite.

After settling, the fate of the P-rich Fe-oxides and vivianite particles would be dictated by the prevailing conditions in the diagenetic environment. Under sulfate limited conditions P removed from the water column is likely to be retained in the sediment, throttling productivity via a negative feedback (März et al., 2008b). When sulfate is available, however, reductive dissolution of Fe-oxides will liberate $PO_4$ to the pore water which may be lost or retained dependent on the locus of dissolution and the availability of other important dissolved species ($Ca^{2+}$, $Fe^{2+}$, $Mg^{2+}$, $F^-$) and reductants (e.g., organic matter, $CH_4$).

Applying these principles to our understanding of Earth History is somewhat speculative but nonetheless interesting, with broad implications for P-cycling. In the largely low-sulfate ferruginous oceans prior to the Great Oxidation Event (GOE; Lyons et al., 2014; Luo et al., 2016), the development of euxinia would have been scarce, enhancing $PO_4$ shuttling to the sediment. If this $PO_4$ was efficiently retained on a global-scale, then $PO_4$ availability in these ancient oceans would have been low, especially considering the reduction in oxidative weathering (Reinhard et al., 2017). By contrast, throughout Earth's history

euxinic marine environments have become more prevalent—a consequence of rising sulfate concentrations (Poulton, 2017). As discussed, euxinic conditions are thought to promote P recycling and its return to the water column; whereas P is believed to be more recalcitrant in ferruginous environments. In a simple sense, therefore, euxinic conditions are touted to be quasi self-sustaining (via a productivity feedback) whereas ferruginous conditions are not. In a wider Earth System sense, however, these gross generalizations may not be valid, and warrant further investigation. Sustained euxinia, for example, could lower sulfate

inventories favoring the development of ferruginous conditions if the $S/Fe_{HR}$ ratio drops below 1.8 (e.g., Poulton and Canfield., 2011). Furthermore, unless reactive Fe enrichments are solely derived from hydrothermal emanations, simple mass balance constraints dictate that sedimentary Fe enrichments are unlikely to occur throughout an anoxic ocean and the source region must be depleted in reactive Fe (e.g., Poulton and Canfield., 2011). Accordingly, to understand the global effects of oxygen

deficiency on P-cycling we must more completely understand the local controls on P-cycling, and how oxygen deficiency developed on an event-by-event basis, to successfully integrate these processes in an Earth system model.

Interestingly, within the present-day oceans, sulfate-driven AOM is known to almost entirely consume methane, serving as an efficient sedimentary methane filter, and preventing this potent greenhouse gas from reaching the atmosphere (Egger et al., 2018; Knittel and Boetius, 2009). In the low sulfate pre-GOE oceans, however, sulfate scarcity may have limited methane oxidation, potentially allowing it to accumulate in the atmosphere (Habicht et al., 2002; Izon et al., 2015, 2017; Kasting et al., 2001). Against this sulfate-lean backdrop, the prevalence of Fe oxide-rich deposits known as banded iron formations (BIFs) dating from the Archean and early Paleoproterozoic (Klein, 2005; Konhauser et al., 2002), are testament to the ubiquity of Fe within the Earth's early oceans. It is possible, therefore, that Fe-AOM was a much more important process within Earth's oxygen deficient early oceans relative to their contemporary counterparts. If true, the interplay between Fe and methane, especially in the prelude of the GOE (e.g., Izon et al., 2017), could have modulated the methane efflux from the ocean to the atmosphere (Riedinger et al., 2014).

## 6 Conclusions

Combining bulk geochemical extractions with microscopic and spectroscopic analyses of handpicked mineral aggregates, we show that vivianite formation occurs within the iron- and methane-rich sediments from the Taixinan Basin, South China Sea. The identified vivianite aggregates are enriched in magnesium but are depleted in manganese, reflecting their growth environment. We argue that as the source of $Fe^{2+}$ for vivianite authigenesis, Fe reduction below the SMTZ was probably driven by Fe-mediated anaerobic oxidation of methane, which has broad implications for Fe and P cycling at depth. Importantly, vivianite authigenesis appears to be restricted to sediments below the SMTZ, where it accounts for ~79% of the Fe-bound P and ~10% of the total sub-SMTZ P burial, respectively. Notably, we calculate that vivianite may account for almost half of the total reactive Fe burial below the SMTZ at Site 973-4. Geochemical conditions that characterize site 973-4 are not uncommon and typify continental margins (Kasten et al., 1998; März et al., 2008a; Riedinger et al., 2014). Thus, vivianite may be an important burial phase for P and Fe below the SMTZ in present-day marine systems (e.g. März et al., 2018). We further speculate that authigenic vivianite might have served as a significant P sink during episodes of marine oxygen deficiency (e.g., März et al., 2008b), throttling productivity via a negative feedback. Moreover, the interplay between Fe and methane, especially in the prelude of the GOE (e.g., Izon et al., 2017), could have curtailed marine methane fluxes with ramifications for the climate system (Riedinger et al., 2014). Thus, a better understanding of vivianite authigenesis is paramount to test long-standing hypotheses linking climate, atmospheric chemistry and the evolution of the biosphere.

## Acknowledgements

Qi Lin is acknowledged for his insight into the study area; whereas formative discussions with Bo Barker Jørgensen, Samantha Joye and Alexandra V. Turchyn helped shape the preparation of this manuscript. We recognize technical assistance from Xiaoping Liao, Wanjun Lu, Chao Li, Caixiang Zhang, Xinna Chai, Jishun Yu, Zihu Zhang and Muhui Zhang at CUG, Wuhan. Daidai Wu, Jie Zhang and Wenjia Ou graciously shared data. The Guangzhou Marine Geological Survey, along with the crew and scientists on board R/V *Ocean VI* are acknowledged for sampling and logistical support. This research was funded by State Key R&D Project of China (Grant 2016YFA0601102), National Natural Science Foundation of China (Grants 41772091, 41472085 and 41802025) and China National Gas Hydrate Project (Grant DD20160211). JL acknowledges financial support via the international exchange program at the School of Earth Sciences, CUG. GI gratefully recognizes support from the Simons Foundation, who funded his contribution under the auspices of the Simons Collaboration on the Origin of Life. Travel support from CUG (Wuhan and Beijing) initiated this collaboration and ignited GI's interest in vivianite. Editorial handling by Tina Treude and reviews by Christian März and an anonymous reviewer are gratefully acknowledged: their expertise, insight and rigor have undoubtedly improved the clarity and quality of the final manuscript.

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

**Table**

**Table 1: Concentrations of specific Fe phases and their ratio to both total reactive Fe and total Fe at Site 973-4. Concentrations of Fe carbonates are taken from Zhang et al. (2018b).**

| Fe burial phase | Concentration below SMTZ ($\mu$mol g$^{-1}$) | Concentration in SMTZ ($\mu$mol g$^{-1}$) | Fe (II)/total reactive Fe below SMTZ | Fe (II)/total reactive Fe in SMTZ | Fe (II) /total Fe below SMTZ | Fe (II) /total Fe in SMTZ |
|---|---|---|---|---|---|---|
| Vivianite | 129 | N.A. | 41% | N.A. | | |
| Pyrite | 4 | 177 | 1% | 52% | | |
| Fe carbonates | 111 | 54 | 36% | 16% | 34% | 32% |
| Total reactive Fe | 311 | 341 | | | | |

**Figures**

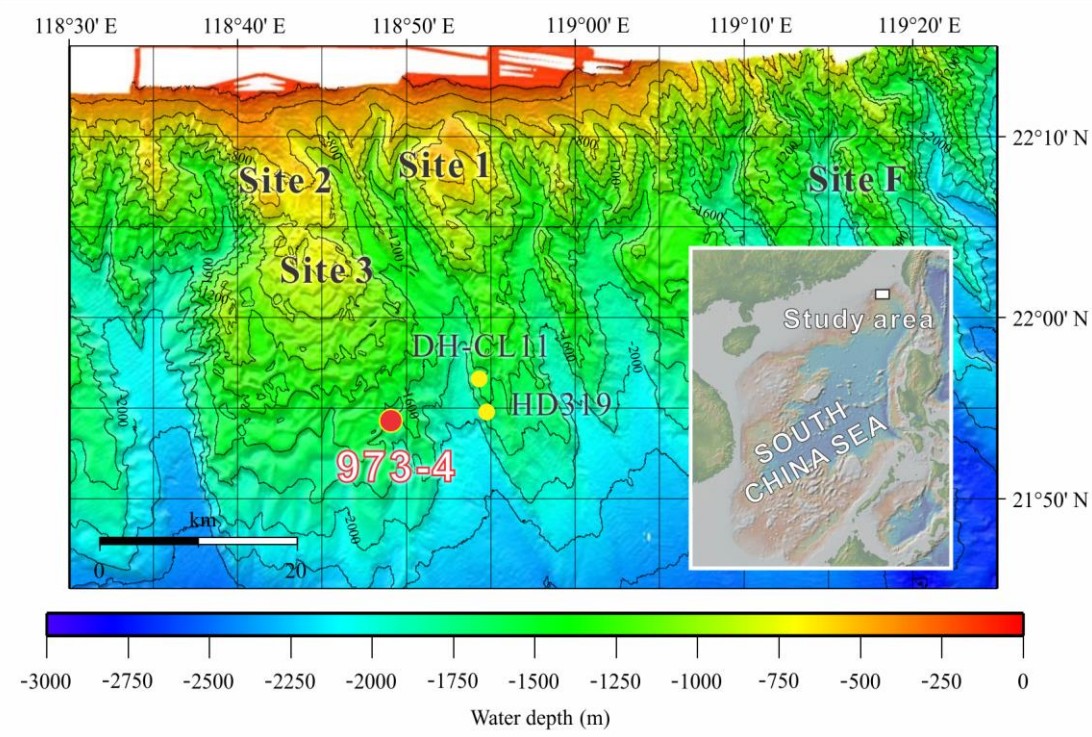

Figure 1: Bathymetric map locating Site 973-4 (red circle) with a geographical insert showing the Taixinan Basin within the wider South China Sea (after Suess et al., 2005).

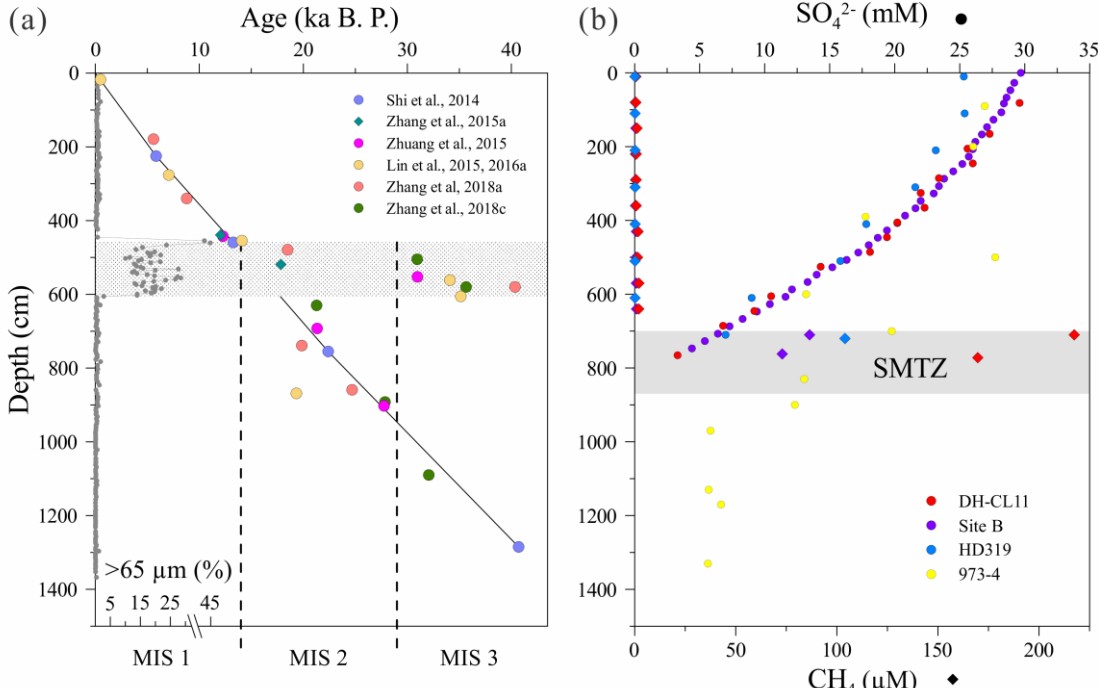

**Figure 2: (a) Stratigraphic distribution of the coarse fraction (> 65 μm, Lin et al., 2017a) and radiocarbon- or δ$^{18}$O-derived age datums from Site 973-4. (b) Profiles of interstitial sulfate (circles) and methane (diamonds) concentrations from Site 973-4 (Zhang et al., 2014) and other adjacent Sites (Lin et al., 2017b; Lu et al., 2012; Ye et al., 2016). Age constraints are either AMS $^{14}$C dates derived from planktonic foraminifera (circles) in calendar years B.P. (Lin et al., 2015, 2016a; Shi et al., 2014; Zhang et al., 2018a; Zhang et al., 2018c; Zhuang et al., 2015) or via correlation to the LR04 benthic oxygen isotope stack (diamonds; Zhang et al., 2015a). MIS—Marine isotope stage. The headspace methane concentrations from Site B and HD 319 were calculated assuming the density of the wet sediments was 1.7 g cm$^{-3}$ (Tenzer and Gladkikh, 2014). The horizontal bars in a and b represent either the coarse layer or the SMTZ, respectively.**

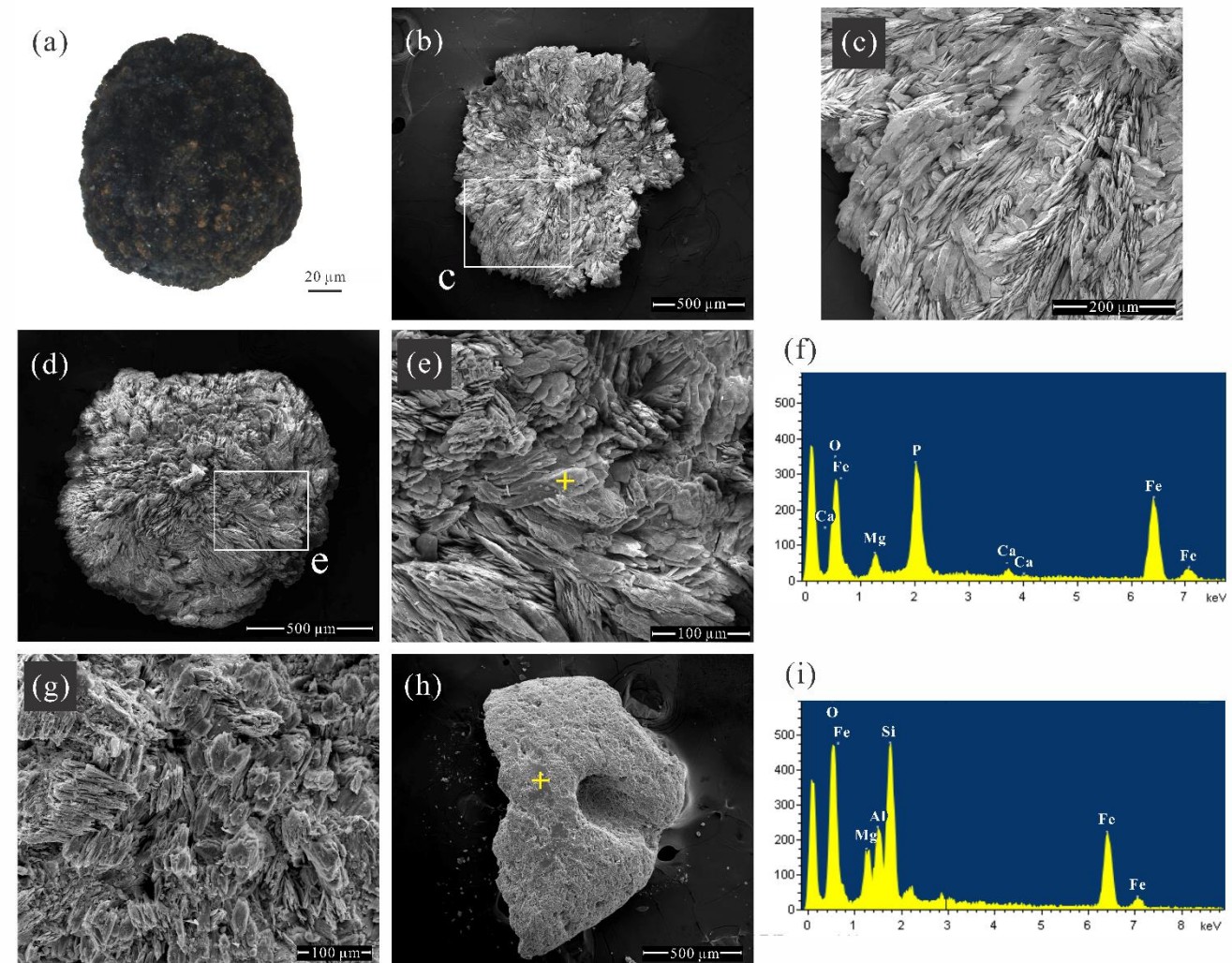

**Figure 3: Images and chemical analyses of handpicked mineral aggregates from core 973-4. (a) Optical photomicrograph of an opaque dark mineral aggregate. (b−e, g) Scanning electron photomicrographs of the blue to black mineral aggregates found beneath the SMTZ. Samples b, d, g were handpicked from sediments at 1087.5, 1093.5, and 973.5 cm depth, respectively. The white boxes in b, d correspond to the fields of view enlarged c and e, respectively. (f) A spot-measurement EDS spectrum corresponding to the yellow cross in e. (h) Scanning electron photomicrograph of the grey to green mineral aggregates from sediments at 602.5 cm depth. (i) A spot-measurement EDS spectrum corresponding to the yellow cross in h.**

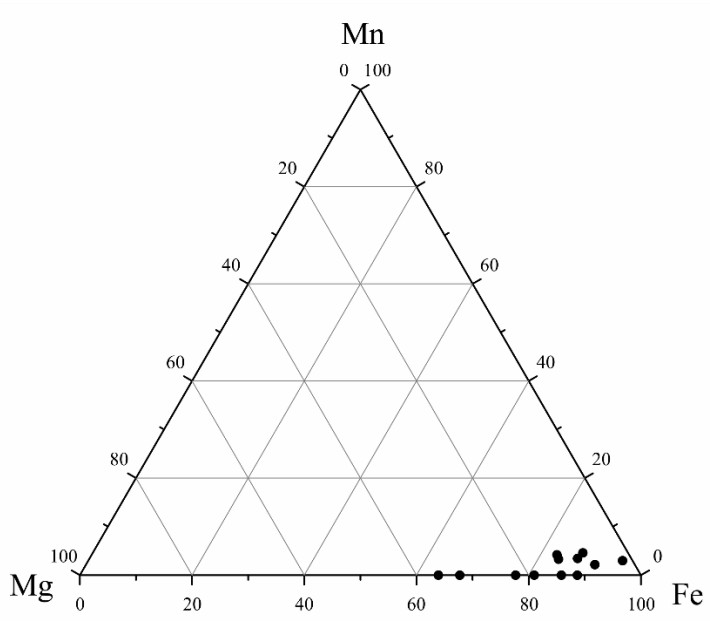

**Figure 4: Ternary plot of the Fe–Mg–Mn contents (in wt.%) of the blue to black mineral aggregates isolated from sediments below the SMTZ. These data were derived from multiple EDS measurements.**

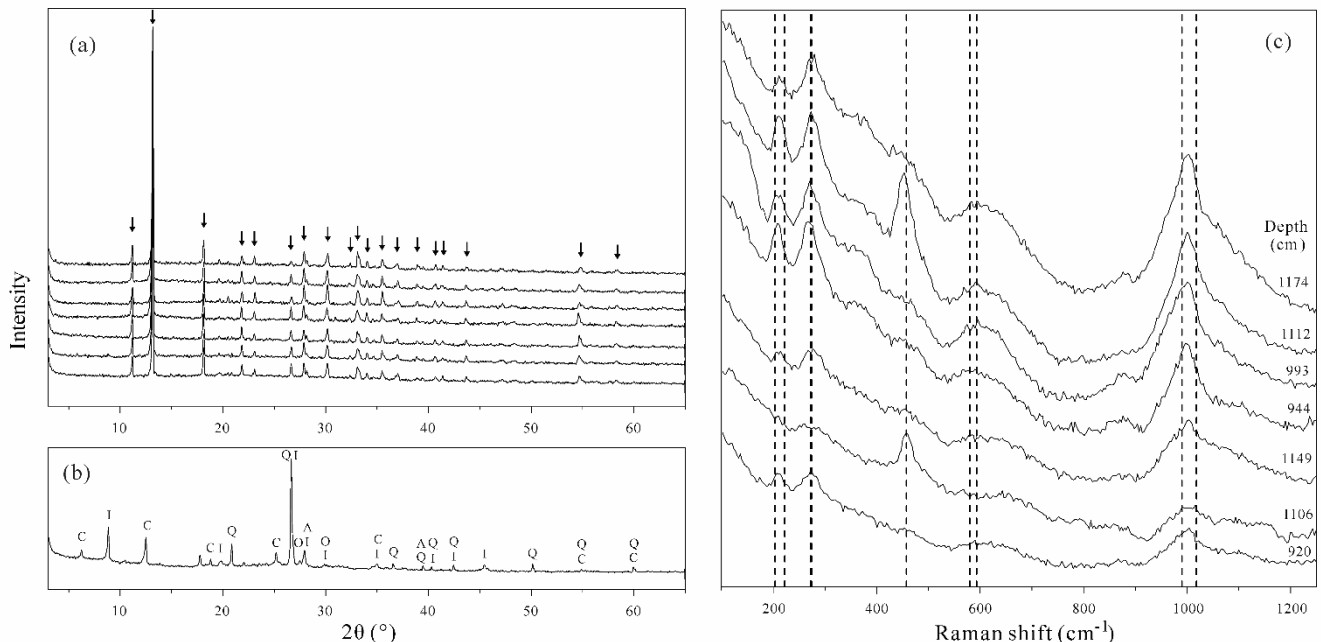

**Figure 5: X-ray diffraction and Raman spectra of handpicked mineral aggregates. (a) XRD spectra of the blue to black mineral aggregates isolated from sediments from below the SMTZ. The spectra of samples from the top to the bottom correspond to samples taken at 920, 944, 993, 1106, 1112, 1149, and 1175 cm depth, respectively. All the identified peaks highlighted by arrows belong to the vivianite reference spectrum. (b) XRD spectrum of the grey to green mineral aggregates obtained from sediments at 1281 cm depth. I = illite, C = chlorites, Q = quartz, A = albite, O = orthoclase. (c) Raman spectra of the blue to black mineral aggregates from below the SMTZ. The dashed lines represent known vivianite spectral features (Piriou and Poullen, 1984).**

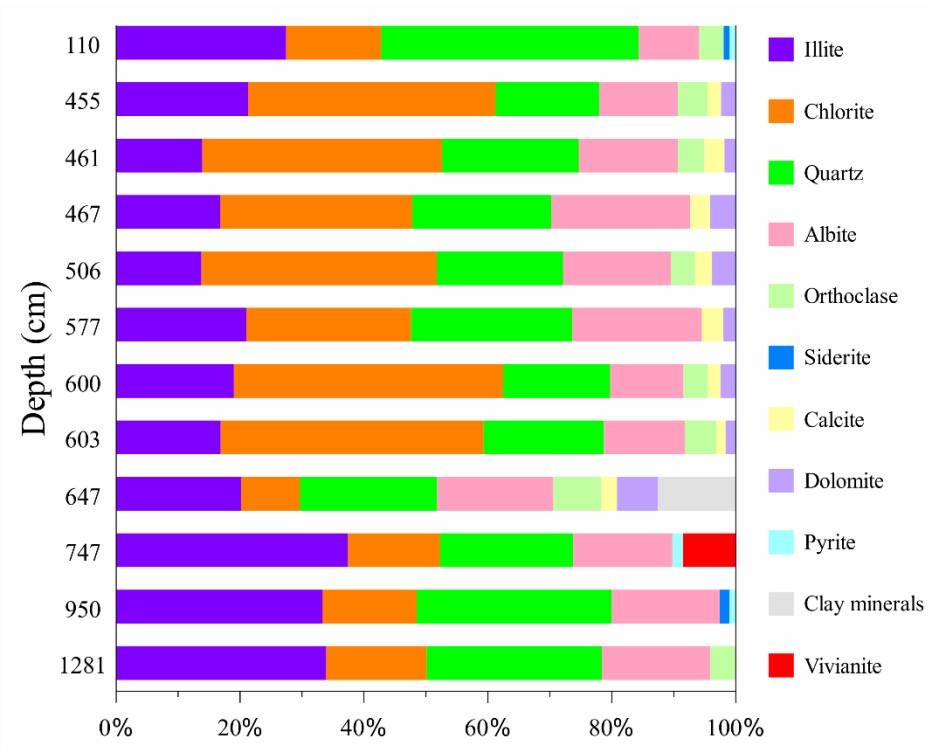

**Figure 6: XRD derived depth profiles depicting the relative mineralogy of the grey to green mineral aggregates isolated from core 973-4. Some of the clay minerals from the sample at 647 cm depth remain unclassified.**

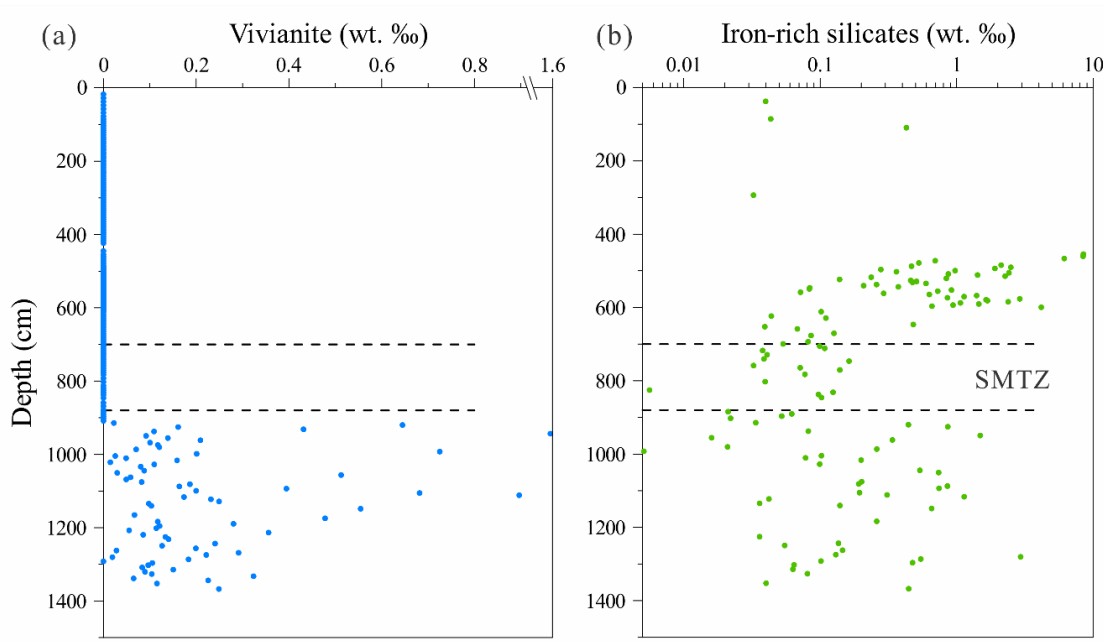

**Figure 7: Depth distribution profiles of handpicked (a) vivianite, (b) Fe-rich silicates. The SMTZ is indicated by horizontal dashed lines.**

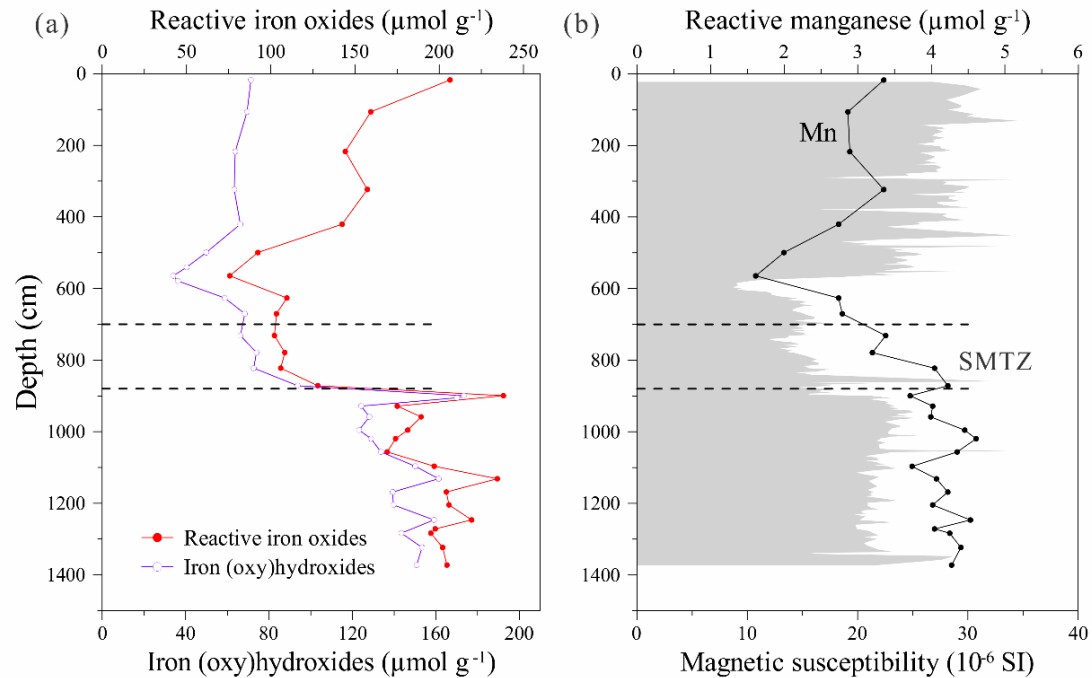

**Figure 8: Depth distribution profiles of (a) Fe (oxy)hydroxides and reactive Fe-oxides, (b) reactive manganese and magnetic susceptibility. Magnetic susceptibility data are adopted from Lin et al. (2017a).**

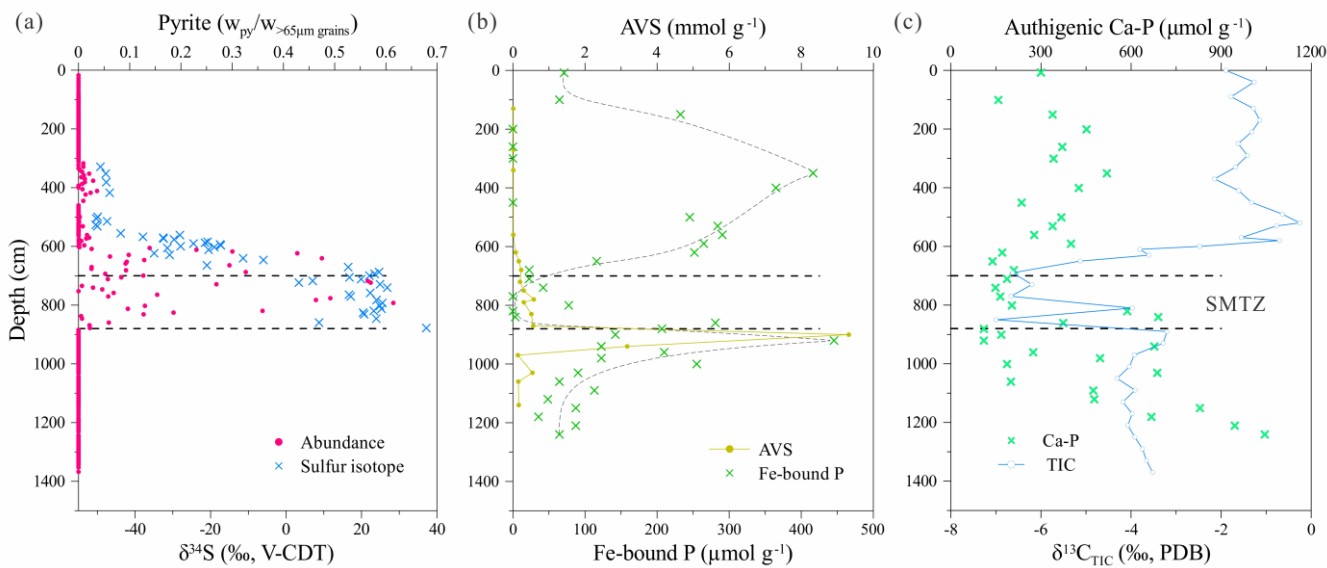

**Figure 9: Depth profiles of (a) handpicked pyrite abundance and sulfur isotopic composition (Lin et al., 2015), (b) acid volatile sulfur (AVS) and Fe-bound P (Zhang et al., 2014, 2018b), (c) authigenic Ca–P and carbon isotopic composition of total inorganic carbon (TIC) (Ou, 2013; Zhang et al., 2018b, 2018c).**

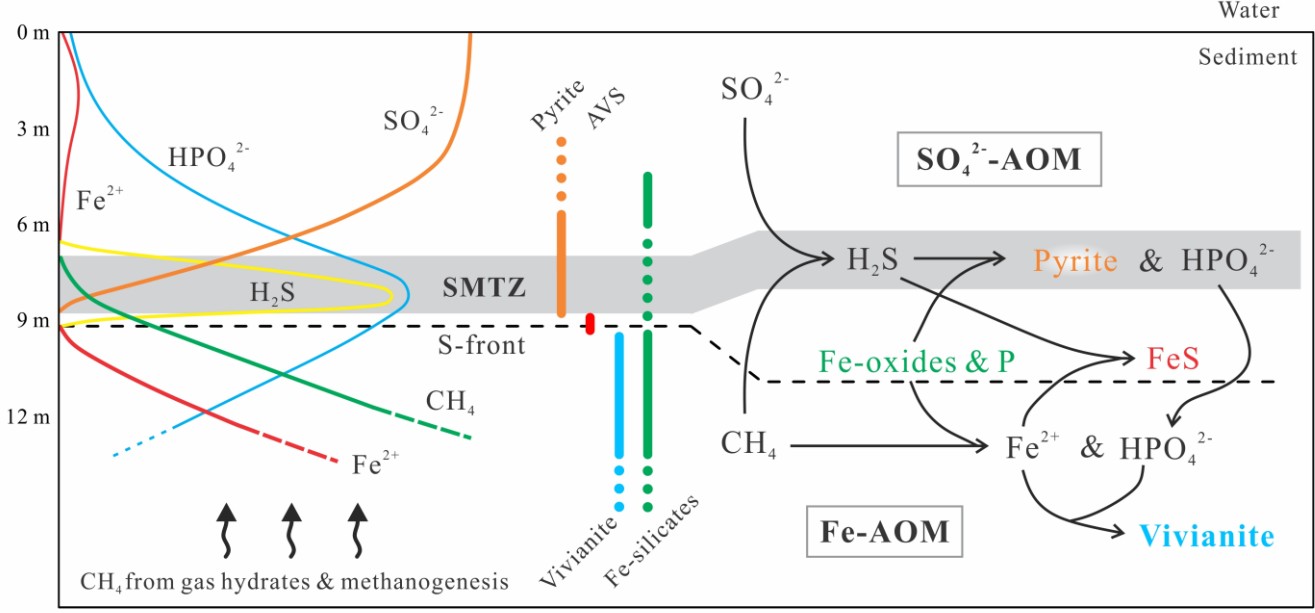

**Figure 10: Schematic representation of vivianite formation below the SMTZ at Site 973-4 in the South China Sea. The left and middle parts illustrate idealized pore water and solid-phase mineral distributions. The right part illustrates the interaction between sulfate- and Fe-driven AOM coupling Fe-S-P-CH₄ cycles in proximity to the SMTZ.**

