# Peer review of "Vivianite formation in methane-rich deep-sea sediments from the South China Sea"

_Biogeosciences, 2018_

## Referee Comment (RC1) · C. März (Referee) · 7 Aug 2018

This manuscript by Liu and co-authors discusses the formation of vivianite in sediments below the SMTZ found in a core from the South China Sea. The manuscript is very well-written and structured, the results support the interpretation, and the evidence for vivianite in this depositional setting is compelling. The topic of the manuscript and the overall quality makes it suitable for publication in Biogeosciences, however, there are a number of issues that the authors need to resolve before the manuscript can be accepted. These issues mostly relate to (a) the description analytical methods and interpretation of the resulting data, (b) the interpretation of the observed geochemical profiles (in particular the suggested downward migration of the SMTZ) in the context of the wider depositional conditions, (c) the downcore record and formation of Ca-P, and

(d) the implications the study might have for specific intervals of the geological past when oceans were anoxic, non-sulphidic (ferruginous).

I hope the authors will find my comment useful. Best regards, Christian März

Sampling of sediments: It is unclear from the text (Methods) how the core sections were processed after sectioning. They were transferred into cool storage. What happened next? Were the sections split on board and samples taken immediately? Were samples stored frozen and/or anoxic to avoid pyrite oxidation prior to freeze-drying? This is crucial for the determination of reactive Fe fractions. Sampling of pore waters: Were pore waters extracted using rhizons? Were they extracted from the closed core by drilling holes into the liner, or were they extracted from the split core surfaces? How long before pore water sampling were the sections split? This is crucial not only for methane and sulphate concentrations, but for any volatile and/or redox sensitive species, i.e., $Fe^{2+}$, $HS^-$, $HPO_4^{2-}$. For example, pore water $Fe^{2+}$ could precipitate as Fe (oxyhydr)oxides and adsorb pore water $HPO_4^{2-}$.

Pore water data: In general, I am not sure if I trust the practise of inferring pore water geochemistry from neighbouring sites, especially since no further information is provided about these. How far away are these nearby sites, and were they affected by the same paleo-depositional processes as the study site? Are there distinct similarities in lithology, sedimentation rates etc that would warrant the "import" of pore water data from these sites? Some of this information can be extracted/inferred from Figure 2 but should be explained in the text as well. Also the methods of pore water sampling and data generation at these nearby sites should be explained.

SMTZ definition: From each of the nearby sites, there are only 1-2 methane data available just below the SMTZ. It should be highlighted in the text that (whether?) this is sufficient to define the SMTZ position. I would also shift the upper boundary of the SMTZ upwards, to the depth where the first methane-free sample was encountered.

Iron extraction: Please cite a reference that defines Fe phases extracted by an anoxic

0.5M HCL solution. I am not actually sure this extraction method has been well-calibrated using different Fe minerals.

Appropriate description of methods: Lots of data are shown (AVS, pyrite, S isotopes, Fe-bound P, Ca-P etc) that are not covered in the Methods section at all. Even if they were published before, a brief account of how these data were generated is required in this manuscript. Were the analyses conducted on splits of exactly the same samples, or nearby samples, or were samples taken at different times? I would also defer from making speculations about what certain data would look like if they had been generated (e.g., pore water $HPO_4^{2-}$).

The Ca-P fraction: A lot of Ca-P is found below the SMTZ, more than Fe-bound P. This is not discussed sufficiently. Is part of the $HPO_4^{2-}$ that is (tentatively) liberated in the SMTZ precipitated as authigenic apatite? Is this supported by pore water Ca and F profiles (if not at this sites, maybe at the nearby sites)? And could the formation of Ca-P be related to the sulphidisation front that consumed part of the Fe-AOM-derived $Fe^{2+}$ and precipitated it as AVS? In other words, does the activity of a sulphidisation front put a constraint on how much Fe(II) phosphate can precipitate below the SMTZ? Are there any estimates of the kinetics of these potentially competing authigenic processes of $Fe^{2+}$ removal from the sub-SMTZ pore waters? This would be very interesting, as it would allow to better link the rate of $HS^-$ production in the SMTZ (which is largely controlled by sulphate diffusion from above and methane delivery from below), the amount of reactive Fe oxides beneath the SMTZ, and the potential to form different authigenic P phases beneath the SMTZ.

"Deepening" of the SMTZ: The suggestion of a previously shallower SMTZ in section 5.1 comes out of the blue, without any specific reasoning for why the authors think this was the case. I do not necessarily disagree with this hypothesis, but the authors need to give some supporting arguments, e.g., maybe the fact that pyrite exists above the SMTZ (although this could also be formed by organoclastic sulphate reduction). They then need to better develop how such a deepening of the SMTZ would have affected all

of the described geochemical parameters, and if observations agree with expectations. Finally, they need to relate the migration to the SMTZ to depositional/environmental processes – if it was not caused by changes in sedimentation rate, was it changes in methane flux from below?

Missing reference: Please cite März et al. (2008, 2018) (both Marine Geology) when stating that a lot of the vivianite is likely finely disseminated. In general, wherever the authors speculate about the wide-spread occurrence of vivianite in methane-rich continental margin sediments, they need to cite the new study by März et al. (2018) (Marine Geology) that comes to exactly the same conclusion.

Vivianite to Ca-P sink switching: When looking at the Ca-P profile, one cannot help but notice that there is an increase to higher values from the SMTZ to the bottom of the core, while Fe-P and vivianite seem to decrease in parallel. This raises the question of the long-term stability of vivianite in the sedimentary record, and its use as paleo SMTZ marker. In fact, vivianite is hardly ever found in older sediments/sedimentary rocks where carbonate fluorapatite is by far the dominant mineral, so there is a strong argument that vivianite is (at least partly) transformed into something else, maybe authigenic apatite?

Importance of findings to the geological past: The link between vivianite formation below the SMTZ in Fe-AOM affected sediments and the potential importance of these processes in the pre-GOE oceans is not well-developed. The actual effects of Fe(II) phosphate formation in an Fe-rich ocean are not discussed at all, only the potential importance of Fe-AOM is mentioned (which is not wrong, but also does not reflect the main story of the research presented here). It would be much more interesting to develop what impact Fe(II) phosphate formation under ferruginous conditions might have had on the marine P cycle. These considerations do not only apply to the pre-GOE ocean; as first proposed by März et al. (2008) (GCA) and further developed by Poulton and Canfield (2011) (Elements), ferruginous conditions existed periodically in the Mesozoic (and probably throughout the Phanerozoic) as well. And as März et

al. (2008) (GCA) pointed out, the precipitation of Fe(II) phosphates occurred under ferruginous conditions during the deposition of Cretaceous black shales on Demerara Rise, sequestering P from the water column and putting a constraint on the anoxia-productivity feedback loop. While we still do not understand enough about the details, and potential effects, of Fe(II) phosphate formation under ferruginous conditions, these earlier studies should be referenced appropriately.

―――――――――――――――――

---

## Referee Comment (RC2) · Anonymous Referee #2 · 25 Aug 2018

Liu et al characterized the distribution and mineralogy of iron-phosphate minerals in methane-rich deep-sea sediments from the South China Sea. They demonstrated the pervasive presence of vivianite authigenesis below the current SMTZ, which is thought to be caused by the downward migration of the former SMTZ where vivianite formation occurred. The data were used to provide insights into Fe-AOM in today's ocean and potentially in the early Earth. The manuscript is definitely suitable for publication in Biogeosciences, systematic and of broad relevance and interest for the geosciences community. I have a few comments for the authors to consider during revision.

Please find below my comments and concerns for the authors to consider during revision:

1). Although other pore-water chemical compositions (PH,PO43-,Ca2+ etc.) is lacking,

a simple geochemical calculation on what's kind of minerals (FeS, CFA, $Mg_3(PO_4)_2$, $Fe_3(PO_4)_2$) are expected to precipitated in/above/under SMTZ can make the P-cycle in methane-rich environments more clearer.

2). Why vivianite is only observed/common under SMTZ (Fig 7a) instead of forming in/above SMTZ. What's the concentration of $HPO_4^{3-}$ are expected when the Mg-rich vivianite is observed (Fig. 4) (The KSP of $Mg_3(PO_4)_2$ are three orders of magnitude higher than $Ca_3(PO_4)_2$, but the $[Mg^{2+}]$ in pore-water are one orders of magnitude higher than $[Ca^{2+}]$). Vivianite is lack in ancient record. Would the vivianite formed here be convert to $Ca_3(PO_4)_2$ further or what's kind of condition where the vivianite can be preserved in sedimentary record and further served as an proxy for methane-rich environment or Fe-AOM activity? In Fig 6, vivianite is only observed in the depth of 747 m? Why there is no vivianite below the 900 cm depth according to the XRF result? There is an inconsistent of vivianite content between the XRF and handpick method.

3). Another concern is the recognition of the current and previous SMTZ. Either pore-water or sediment has its own validity in revealing the characteristics and mechanisms of seepage. For example, the geochemical data obtained from the solid fraction of sediments and from authigenic carbonates provide time-averaged information on bio-geochemical processes on a timescale of years to centuries. Sediment pore waters and seep-dwelling fauna, on the other hand, provide information on much shorter timescales, spanning from days to months. This issue need to be considered and discussed in discussion.

4). Along with 3), one needs to mention the nature of the seeps. It is well known that seeps are heterogeneous both in time and space. I would find interesting that they describe in a few words the inherent nature of seeps in the introduction section and consequently highlight their findings in the discussion section. The shift of former and current SMTZ is exactly caused by the varying of flux of fluids.

---

## Author Comment (AC1) · 22 Sep 2018

We thank Dr. März for his comments and insight. His requests have helped to clarify the text and highlight omissions from our initial submission. We reply to each of his comments in turn and aim to revise the manuscript accordingly.

R1.1). Comment (sampling of sediments): It is unclear from the text (Methods) how the core sections were processed after sectioning. They were transferred into cool storage. What happened next? Were the sections split on board and samples taken immediately? Were samples stored frozen and/or anoxic to avoid pyrite oxidation prior to freeze-drying? This is crucial for the determination of reactive Fe fractions. Sampling of pore waters: Were pore waters extracted using rhizons? Were they extracted from

the closed core by drilling holes into the liner, or were they extracted from the split core surfaces? How long before pore water sampling were the sections split? This is crucial not only for methane and sulphate concentrations, but for any volatile and/or redox sensitive species, i.e., Fe2+, HS-, HPO42-. For example, pore water Fe2+ could precipitate as Fe (oxyhydr)oxides and adsorb pore water HPO42-.

Reply: We acknowledge this oversight and agree that more detail should be included to clarify how the sediments/pore waters were handled and collected. We intend to clarify this in the amended manuscript.

After sectioning on board, the cores were stored below 4 °C. Half-cores were then split and subsampled immediately after the cruise. Sub-samples were stored frozen ($-20$ °C) prior to freeze-drying. The obvious peak of Fe-oxides at ~900 cm depth (Fig. 8a) may include part of acid volatile sulfur (AVS) pool since AVS oxidizes to dithionite- and oxalate-extractable Fe phases during freeze-drying or exposure to air. It is difficult, however, to quantify the influence of pyrite oxidation on the reactive Fe distributions at Site 973-4.

Unfortunately, the pore waters were not immediately extracted; instead, they were extracted several months later via centrifugation of previously frozen sub-samples. Consequently, the pore water chemistry of Site 973-4 (Fig. 2b) is more likely to reflect post-recovery oxidation or contamination rather than sediment-housed processes. For example, volatile sulfide lost and/or sulfate distributions could be modified by sulfide oxidation. Alternatively, as R1 suggests, precipitation of dissolved Fe could influence the distributions of Fe-sensitive species. It is for this reason that we opted to deemphasize the importance of the published pore water chemistry in favor of proximal published data (Fig. 2b; see R1.2).

R1.2). Comment (pore water data and SMTZ definition): In general, I am not sure if I trust the practise of inferring pore water geochemistry from neighbouring sites, especially since no further information is provided about these. How far away are these

nearby sites, and were they affected by the same paleo-depositional processes as the study site? Are there distinct similarities in lithology, sedimentation rates etc that would warrant the "import" of pore water data from these sites? Some of this information can be extracted/inferred from Figure 2 but should be explained in the text as well. From each of the nearby sites, there are only 1-2 methane data available just below the SMTZ. It should be highlighted in the text that (whether?) this is sufficient to define the SMTZ position. I would also shift the upper boundary of the SMTZ upwards, to the depth where the first methane-free sample was encountered. Also the methods of pore water sampling and data generation at these nearby sites should be explained.

Reply: We agree with R1's concerns but, given the complications concerning the quality of the pore water data at Site 973-4 (See R1.1), we believe that synthesizing pore water data from the surrounding sites is likely to be more representative. We stress that we do not advocate this approach to replace pore water analysis but, in its absence, we argue that pore water data from nearby sites when combined with solid-phase distributions from Site 973-4 allow to estimate an approximate position of the SMTZ that satisfies for the purposes of our manuscript.

Sites from which we obtained pore water data are separated from Site 973-4 by a few kilometers. These sites are lithologically similar and dominated by silty clay. Unfortunately, age constraints and sedimentation rates are lacking at the other sites. The pore water chemistry from the combined sites, however, is analogous, indicating that each site was likely affected by the same depositional process(es). Solid phase distributions at Site 973-4 support this inference. At the sulfidization front, sulfide is known to diffuse out of the SMTZ promoting reductive dissolution of Fe-oxides and the sequestration of elemental sulfur and Fe monosulfides ($3H_2S + 2FeOOH \rightarrow S^0 + 2FeS + 4H_2O$). At Site 973-4 there are two pronounced peaks of elemental sulfur at 730 and 880 cm depth, while AVS concentrations reach their maximum at $\sim$900 cm depth (Fig. 9b). Accordingly, we combine these site-specific solid-phase records, with the "imported" pore water data to infer the SMTZ is currently around 700–880 cm depth at Site 973-4.

As R1 suggested, however, the first methane-free samples appear at 640 cm depth right above the SMTZ. Considering the first elemental sulfur peak at 730 cm depth, we stress that it is more appropriate to define the upper boundary of the SMTZ as a rough depth for the purposes of our manuscript, i.e. around 700 cm depth.

Given following discussion, and the need to draw on the data of others, we agree with R1 that the pertinent details of pore water sampling and analysis should also be included in the manuscript. Clarifying text to rectify this oversight is: Pore water extraction from Site DH-CL11 was conducted on shore via centrifugation, exploiting sample aliquots that had been immediately taken and stored under vacuum at $-80$ °C (Lin et al., 2017). Pore water samples from Site B and Site HD319 were collected immediately after recovery by vacuum extraction (Lu et al., 2012; Ye et al., 2016). The extracted pore water at these three sites was preserved in cryogenic vials at 4 °C prior to sulfate analysis by ion chromatography. Methane concentrations at these three sites were determined via sediment plug sampling and gas chromatographic analysis of the resultant headspace gas (Lin et al., 2017; Lu et al., 2012; Ye et al., 2016).

R1.3). Comment (iron extraction): Please cite a reference that defines Fe phases extracted by an anoxic 0.5M HCL solution. I am not actually sure this extraction method has been well calibrated using different Fe minerals.

Reply: The extraction method with anoxic 0.5 M HCl solution was adopted from Holmkvist et al. 2011 and 2014 (GCA). More than 97% of amorphous iron (hydro)oxides and ferrihydrite can be extracted, while other Fe-oxides minerals are apparently recalcitrant to this extractant (Wallmann et al., 1993, Limnology and Oceanography; Haese et al., 1997, GCA). Compared to the more well-known dithionite method (e.g. Poulton and Canfield, 2005), the HCl method has not been well calibrated. Therefore, we will ensure the relevant references are included in the manuscript but intend to refer dithionite-extractable Fe as Fe-oxides. These data will be used for the ensuing discussion.

R1.4). Comment (appropriate description of methods): Lots of data are shown (AVS, pyrite, S isotopes, Fe-bound P, Ca-P etc) that are not covered in the Methods section at all. Even if they were published before, a brief account of how these data were generated is required in this manuscript. Were the analyses conducted on splits of exactly the same samples, or nearby samples, or were samples taken at different times? I would also defer from making speculations about what certain data would look like if they had been generated (e.g., pore water HPO42-).

Reply: Following R1's advice, a brief description of how the various published datasets were generated will be included in the amended manuscript. This will be appended to the existing methods section. The clarifying text requested by R1 is: Additional published data (e.g., Fig. 9) was determined following established protocols and full methodological details are provided in the respective papers. Accordingly, only a brief description is provided here. Acid volatile sulfide was liberated via HCl distillation and trapped by zinc acetate, its concentration was then determined spectrophotometrically (Zhang et al., 2014). Pyrite aggregates were handpicked from the coarse-fraction (> 65 $\mu$m) with its abundance expressed relative to the coarse grains. The sulfur isotopic composition of handpicked pyrite was determined directly using a Delta V Plus isotope ratio mass spectrometer (IRMS; Lin et al., 2015). The solid-phase distribution of P was revealed through the SEDEX sequential extraction scheme (Ruttenberg, 1992). Iron-bound P and Ca-P were extracted by citrate-bicarbonate-dithionite and Na-acetate buffer, respectively (Zhang et al., 2018b). The carbon isotopic composition of total inorganic carbon was determined after treatment with phosphoric acid using a Finnigan MAT-252 IRMS (Ou, 2013). Finally, magnetic susceptibility was investigated using a MFK1-FA kappameter (Lin et al., 2017a).

All analyses were conducted on splits of exactly the same samples from the same core which was obtained from a cruise in 2011. Although test date is different for each analysis, all the sediment samples were kept frozen before analyses. Following R1's advice, speculations will be removed from an amended manuscript.

R1.5). Comment (the Ca-P fraction): A lot of Ca-P is found below the SMTZ, more than Fe-bound P. This is not discussed sufficiently. Is part of the HPO42- that is (tentatively) liberated in the SMTZ precipitated as authigenic apatite? Is this supported by pore water Ca and F profiles (if not at this site, maybe at the nearby sites)? And could the formation of Ca-P be related to the sulphidisation front that consumed part of the Fe-AOM-derived Fe2+ and precipitated it as AVS? In other words, does the activity of a sulphidisation front put a constraint on how much Fe(II) phosphate can precipitate below the SMTZ? Are there any estimates of the kinetics of these potentially competing authigenic processes of Fe2+ removal from the sub-SMTZ pore waters? This would be very interesting, as it would allow to better link the rate of HS- production in the SMTZ (which is largely controlled by sulphate diffusion from above and methane delivery from below), the amount of reactive Fe oxides beneath the SMTZ, and the potential to form different authigenic P phases beneath the SMTZ.

Reply: This is an interesting point, and one that warrants inclusion in the discussion of the amended manuscript, albeit still speculative due to the lack of appropriate data.

As pointed out by R1, authigenic Ca-P is the major P sink at Site 973-4 ($\sim$55%), in agreement with its globally estimated importance. Indeed, Ca-P concentrations are much higher than Fe-bound P throughout the studied core. It is well known that a steady supply of phosphate, fueled by the reductive dissolution of Fe-oxides, fosters thermodynamically favorable conditions capable of promoting apatite authigenesis in proximity to the SMTZ (Egger et al., 2015a, Fig. S8; März et al., 2018). Thus, as suggested by R1, it is possible that some of the SMTZ-sourced phosphate is consumed and precipitated as authigenic apatite, providing the other necessary chemical constituents are available. While there are no reliable pore water data from Site 973-4 (R1.1), dissolved Ca2+ concentrations from Site B are consistent with ongoing apatite authigenesis, with a linear down-core decrease from the sediment-water interface to the upper SMTZ (12.5–2.5 mM) where most of the Ca2+ has been consumed (Ye et al., 2016). To our knowledge, however, there are no available dissolved F− and PO4

data to unequivocally confirm apatite precipitation rather than other calcium-harboring phases.

Vivianite formation below the SMTZ depends on PO4 and Fe2+ availability, which we suggest are mainly supplied from PO4 liberation in the SMTZ and the upward flux of Fe2+ from iron reduction at depth. Kinetics notwithstanding, providing there is sufficient Ca2+ and F− apatite will remove PO4 competing and effectively limiting vivianite precipitation. At Site 973-4 vivianite occurs below ∼900 cm depth suggesting that even if apatite authigenesis is a competing process, Ca2+ and F− might have been consumed at this depth. Importantly, vivianite authigenesis occurs below the sulfidization front (∼900 cm) implying that FeS precipitation is kinetically favored relative to vivianite precipitation right below the SMTZ. We speculate, therefore, that the pore water chemistry and the activity of sulfidization front will influence the depth of vivianite formation. Intensified activity at the sulfidization front would liberate more phosphate, however, the extra sulfide flux may serve to nullify the ascent of Fe2+ from depth, especially as highly reactive iron is consumed, preventing vivianite formation or forcing it to occur in deeper sediment-layers.

Unfortunately, we do not have the data to definitively test the hypothesized competing role of apatite and FeS authigenesis in vivianite precipitation. However, any process that serves to deplete pore water PO4 and Fe2+ will hypothetically compete with vivianite formation. Future work in/around the study area should focus on generating comprehensive pore water and solid phase data sets analyses, coupled with reactive transport modeling, to better link the rate of HS- production in the SMTZ, the amount of reactive Fe-oxides beneath the SMTZ, and the potential to form different authigenic P phases beneath the SMTZ.

R1.6). Comment ("deepening" of the SMTZ): The suggestion of a previously shallower SMTZ in section 5.1 comes out of the blue, without any specific reasoning for why the authors think this was the case. I do not necessarily disagree with this hypothesis, but the authors need to give some supporting arguments, e.g., maybe the fact that pyrite

exists above the SMTZ (although this could also be formed by organoclastic sulphate reduction). They then need to better develop how such a deepening of the SMTZ would have affected all of the described geochemical parameters, and if observations agree with expectations. Finally, they need to relate the migration to the SMTZ to depositional/environmental processes – if it was not caused by changes in sedimentation rate, was it changes in methane flux from below?

Reply: Following R1's advice the following discussion is warranted. If we accept that the current SMTZ is around 700–880 cm depth based on the arguments presented in R1.2, we can speculate on the SMTZ migration at our site. The preservation of Fe-oxides and absence of pyrite below the SMTZ were most likely caused by the rapid upward migration of the SMTZ from below 14 m (i.e. deeper than the core) to the middle of our core. The short exposure of Fe-oxide minerals to sulfidic conditions led to incomplete reduction and sulfidization (März et al., 2018). After migrating, the SMTZ has varied around 560–880 cm, leaving pronounced imprints on the pyrite, Fe-oxide, Fe-silicate, total inorganic carbon and magnetic susceptibility profiles (Figs. 7–9). It should be noted that pyrite formed by organoclastic sulfate reduction usually features more negative $\delta34S$ values ($-50$ per mill) such as those seen at $\sim$300–560 cm depth. Whereas, the $\delta34S$ excursion seen at $\sim$560–880 cm depth is best explained by sulfate driven AOM activity at and above the current SMTZ. Assimilating these observations, we suggest that the SMTZ might have slowly migrated downward from $\sim$560–700 cm to its current position at $\sim$700–880 cm depth.

This migration pattern may explain why we observed vivianite at 747 cm depth (Fig. 6). The paleo-SMTZ may be a little shallower (e.g. $\sim$560–700 cm depth) allowing vivianite precipitation at shallower depths (e.g. 747 cm depth). However, we speculate that most the vivianite formed at/around the current SMTZ was readily converted to Fe-sulfide phases via reaction with H2S during the downward migration of the SMTZ, concentrating vivianite below the current SMTZ. If the SMTZ continued to descend within the sediment, we would expect that the pronounced solid-phase imprints observed at ∼560–880 cm would also descend to deeper sediments below the current SMTZ.

Given that the sedimentation rate at Site 973-4 is almost constant over the duration of the core (Fig. 2a), we attribute the SMTZ migration pattern to a change in the methane flux from depth. During sea-level low stands around the Last Glacial Maximum, the low hydrostatic pressure would destabilize the underlying methane clathrates, enhancing methane fluxes while promoting a rapid upward migration of the SMTZ (Borowski et al., 1996). The subsequent Holocene sea-level rise diminished methane fluxes and might have instigated a slow downward migration of the SMTZ to its current position. Our study further supports recent findings that the SMTZ needs to be fixed at a specific sediment interval (i.e. ∼560–880 cm depth) for thousands of years to allow for the development of significant authigenic Fe(II) phosphate mineral enrichments below the SMTZ (März et al., 2018).

R1.7). Comment (missing reference): Please cite März et al. (2008, 2018) (both Marine Geology) when stating that a lot of the vivianite is likely finely disseminated. In general, wherever the authors speculate about the wide-spread occurrence of vivianite in methane-rich continental margin sediments, they need to cite the new study by März et al. (2018) (Marine Geology) that comes to exactly the same conclusion.

Reply: We apologize for the omission of the reviewer's most recent work. We had submitted our paper before the 2018 study had been formally published. As suggested, in our manuscript, we will cite März et al., 2008 and 2018 to fully credit the importance of finely disseminated vivianite in methane-rich continental margin settings.

R1.8). Comment (vivianite to Ca-P sink switching): When looking at the Ca-P profile, one cannot help but notice that there is an increase to higher values from the SMTZ to the bottom of the core, while Fe-P and vivianite seem to decrease in parallel. This raises the question of the long-term stability of vivianite in the sedimentary record, and its use as paleo SMTZ marker. In fact, vivianite is hardly ever found in older sediments/sedimentary rocks where carbonate fluorapatite is by far the dominant mineral, so there is a strong argument that vivianite is (at least partly) transformed into something else, maybe authigenic apatite?

Reply: The question concerning vivianite's longevity in the sedimentary record is a good one and, indeed, an outstanding question that needs to be addressed more generally. Unfortunately, in the context of the available data, we cannot unequivocally test this hypothesis and thus opt to expand the discussion to encompass the available observations and interpretations. However, given the ambiguity, we also elect to "tone-down" our language to convey uncertainty in the amended manuscript.

The increase in Ca-P concentrations noted by R1 may reflect the conversion of labile P-species into authigenic apatite, so-called sink switching. Sink switching is considered to be the dominant global process of apatite authigenesis in marine sediments (e.g. Ruttenberg and Berner, 1993; Slomp et al., 1996). Whether vivianite is converted to authigenic apatite at depth, while possible, remains an open question. Moreover, if it were a simple conversion then one would predict a linear increase in apatite with a concomitant decrease in vivianite. Closer inspection of the relevant solid phase records reveals that this is not the case and, in fact, the relationship between apatite and vivianite concentrations is ambiguous and largely decoupled at Site 973-4.

Alternatively, the "increase" in authigenic Ca-P from the SMTZ to the bottom of the core may reflect non-vivianite-related apatite authigenesis at depth. Upward fluxes of $Ca^{2+}$ and $F^-$ to the SMTZ have been observed in continental margin settings (e.g. Clemens et al., 2016, IODP 353; März et al., 2018), which would provide the necessary chemical constituents to promote sub-SMTZ apatite authigenesis. As discussed in R1.5, if apatite is a more favorable phosphate sink, then precipitation of this mineral phase could hypothetically curtail vivianite authigenesis. In this scenario, although the two phosphate phases are linked, they are linked through $Ca^{2+}$ (and $F^-$) availability rather than stability and conversion of one phase to another.

The apparent decrease in vivianite with depth could also be completely decoupled from apatite authigenesis. For example, vivianite is sensitive to oxic conditions, thus it is possible that vivianite in deep-time records was oxidized to Fe-oxides (e.g. hematite; Berner, 1981), Fe(III) phosphate (e.g. koninckite; März et al., 2008, Marine Geology) or some other unknown phases during tectonic uplift. It provides another clue for future research that vivianite may be partly preserved as pseudomorphs (e.g. metavivianite) or with special markers in P-bearing sedimentary rocks. However, the perceived absence of vivianite in aged sedimentary archives could equally be an artifact of insufficient surveys. Our work, as well as that of R1 (März et al., 2018), has shown that vivianite is likely a finely disseminated phase and thus may have escaped detection in many studies. Moreover, typically employed extraction schemes co-extract vivianite, meaning that Fe-P in deep time records could have been wrongly ascribed, and could be vivianite (c.f., März et al., 2008, GCA).

Future work, targeting long cores coupling aqueous- and solid-phase analysis, is required to definitively test our hypotheses linking vivianite authigenesis and Fe-AOM activity, as well as to examine the long-term stability of vivianite in the sedimentary record. More routine application of XRD and modification of solid-phase extraction protocols is also necessary (pers. comm. S. Poulton, 2018) to answer questions relating to the geological significance of vivianite.

R1.9). Comment (importance of findings to the geological past): The link between vivianite formation below the SMTZ in Fe-AOM affected sediments and the potential importance of these processes in the pre-GOE oceans is not well-developed. The actual effects of Fe(II) phosphate formation in an Fe-rich ocean are not discussed at all, only the potential importance of Fe-AOM is mentioned (which is not wrong, but also does not reflect the main story of the research presented here). It would be much more interesting to develop what impact Fe(II) phosphate formation under ferruginous conditions might have had on the marine P cycle. These considerations do not only apply to the pre GOE ocean; as first proposed by März et al. (2008) (GCA) and further developed

by Poulton and Canfield (2011) (Elements), ferruginous conditions existed periodically in the Mesozoic (and probably throughout the Phanerozoic) as well. And as März et al. (2008) (GCA) pointed out, the precipitation of Fe(II) phosphates occurred under ferruginous conditions during the deposition of Cretaceous black shales on Demerara Rise, sequestering P from the water column and putting a constraint on the anoxia productivity feedback loop. While we still do not understand enough about the details, and potential effects, of Fe(II) phosphate formation under ferruginous conditions, these earlier studies should be referenced appropriately.

Reply: These are interesting and valid points. As detailed by R1, the seawater chemistry was vastly different between ancient oceans and their modern counterparts. In our original manuscript, we specifically referred to the predominantly ferruginous state of pre-GOE oceans, yet we concede that ferruginous conditions were also a periodic feature post-GOE oceans (März et al., 2008; Poulton and Canfield, 2011). The latter scenario is subtly different from the former, however, due to ingrowth of the seawater sulfate reservoir (Canfield et al., 2000). Accordingly, we welcome the opportunity to extend the discussion, fortifying the stance of others while more completely developing our hypotheses concerning the importance of vivianite, and its wider role in P-cycling in ancient oceans.

Where basal waters are oxygenated, we expect a similar pore water pattern to that observed at Site 973-4 or other modern open marine settings (Fig. 10), where vivianite is dominantly formed below the SMTZ. If oxygen concentrations drop, however, vivianite authigenesis is expected to be different with implications for P-cycling and productivity. These intricacies, however, remain to be fully explored.

Under ferruginous (anoxic and $Fe^{2+}$-rich) conditions dissolved phosphate would be adsorbed to and/or co-precipitated with Fe-oxides (März et al., 2008), or even precipitated as vivianite directly if $Fe^{2+}$ and $PO_4$ concentrations were sufficiently high (Dijkstra et al., 2018b). This P shuttle is (at least) partially eradicated if euxinic conditions develop in the water column caused by sulfidization of Fe-oxides and/or vivianite presumably
governed by particle settling kinetics. After settling, the fate of the P-rich Fe-oxides and vivianite particles would be dictated by the prevailing conditions in the diagenetic environment. Under sulfate limited conditions P removed from the water column is likely to be retained in the sediment-pile, throttling productivity via a negative feedback (März et al., 2008). When sulfate is available, however, reductive dissolution of Fe-oxides will liberate phosphate to the pore water which may be lost or retained dependent on the locus of dissolution and the availability of other important dissolved species (Ca2+, Fe2+, Mg2+, F-) and reductants (e.g., OM, CH4).

Applying these principles to our understanding of Earth History is somewhat speculative but nonetheless interesting, with broad implications for P-cycling. In the largely low-sulfate ferruginous oceans prior to the GOE, the development of euxinia would have been scarce, enhancing phosphate shuttling to the sediment. If this phosphate was efficiently retained on a global-scale, then phosphate availability in these ancient oceans would have been low, especially considering the reduction in oxidative weathering (Reinhard et al., 2017). By contrast, throughout Earth's history euxinic marine environments have become more prevalent – a consequence of rising sulfate concentrations. As discussed, euxinic conditions promote P recycling and its return to the water column; whereas P is more recalcitrant in ferruginous environments. In a simple sense, therefore, euxinic conditions are touted to be quasi self-sustaining (via a productivity feedback) whereas ferruginous conditions are not. In a wider Earth System sense, however, these gross generalizations may not be valid, and warrant further investigation. Sustained euxinia, for example, could lower sulfate inventories causing the development of ferruginous conditions if S/FeHR ratios drop below 2 (e.g., Poulton and Canfield., 2011). Furthermore, unless reactive iron enrichments are solely derived from hydrothermal emanations, simple mass balance constraints dictate that sedimentary iron enrichments are unlikely to occur throughout an anoxic ocean and the source region must be depleted in reactive iron. Consequently, to understand the global effects of oxygen deficiency on P-cycling we must more completely understand the local controls on P-cycling and how oxygen deficiency developed on an event-by-event basis

to successfully integrate these processes in an Earth system model.

Sincerely, on behalf of all authors,

Jiarui Liu

---

## Author Comment (AC2) · 22 Sep 2018

We would like to express our gratitude to R2. Most of the points raised here were also raised by R1, we have attempted to cross reference between our responses to prevent repetition. We reply to each of the reviewer's comments in turn and aim to revise the manuscript accordingly.

R2.1). Comment: Although other pore-water chemical compositions (PH,PO43-,Ca2+ etc.) is lacking, a simple geochemical calculation on what's kind of minerals (FeS, CFA, Mg3(PO4)2, Fe3(PO4)2) are expected to precipitated in/above/under SMTZ can make the P-cycle in methane-rich environments more clearer.

Reply: Pore water chemistry is essential to conduct a comprehensive series of calcu-
lations on mineral precipitation in marine sediments. As discussed in R1.1, we lack robust pore water data and thus we draw on PHREEQC calculations of pore water saturation indexes from other sites (Egger et al., 2015a, Baltic Sea, Fig. S8; März et al., 2018, open marine margins) to address this comment.

Hydroxyapatite saturation indexes above 0 suggest that the steady supply of phosphate from reductive dissolution of Fe-oxides creates thermodynamically favorable conditions for carbonate fluorapatite (CFA) precipitation around the SMTZ, whereas CFA formation appears to be a minor sink for pore water phosphate below the SMTZ (März et al., 2018). Higher saturation indexes for vivianite are typically obtained below the SMTZ. Here, the upward flux of $Fe_{2+}$ from iron reduction and downward flux of phosphate from the reductive dissolution of Fe-oxides create favorable geochemical conditions for vivianite precipitation below the SMTZ, consistent with the vivianite distribution observed at Site 973-4. The solubility product constant pKsp ($-logKsp$) of Mg phosphate is 24, while the pKsp of vivianite is 36 (Nriagu, 1972), making the former mineral more soluble. Moreover, Mg concentrations are typically up to three orders of magnitude higher than those of Fe in marine sediments (e.g. Hu et al., 2015, Site D-5). Consequently, Mg phosphate precipitation is unlikely. Rather, $Mg_{2+}$ is more likely to be co-precipitated with vivianite, forming Mg-rich vivianite below the SMTZ.

Besides phosphate minerals, Fe sulfides are enriched in and above the SMTZ. Pyrite is a common mineral in marine sediments where dissolved sulfide, produced by either organoclastic sulfate reduction (above the SMTZ) or sulfate driven anaerobic oxidation of methane (within the SMTZ), is trapped by Fe-oxides, forming Fe monosulfide. Fe monosulfides are gradually converted to pyrite as a permanent sink for sulfur. The acid volatile sulfur peak at ∼900 cm depth contains large amounts of Fe monosulfides, and is referred to as the sulfidization front (S-front). The dissolved Fe is sourced either from the reductive dissolution of Fe-oxides by sulfide at the same depth or iron reduction at depth. Consequently, the formation of Fe sulfide is modulated by sulfate penetration depth.

R2.2). Comment: Why vivianite is only observed/common under SMTZ (Fig 7a) instead of forming in/above SMTZ. What's the concentration of HPO43- are expected when the Mg-rich vivianite is observed (Fig. 4) (The KSP of Mg3(PO4)2 are three orders of magnitude higher than Ca3(PO4)2, but the [Mg2+] in pore-water are one orders of magnitude higher than [Ca2+]). Vivianite is lack in ancient record. Would the vivianite formed here be convert to Ca3(PO4)2 further or what's kind of condition where the vivianite can be preserved in sedimentary record and further served as a proxy for methane rich environment or Fe-AOM activity? In Fig 6, vivianite is only observed in the depth of 747 m? Why there is no vivianite below the 900 cm depth according to the XRF result? There is an inconsistent of vivianite content between the XRF and handpick method.

Reply: We did not identify any macro-vivianite within or above the SMTZ. Generally, vivianite is unstable in the presence of sulfide that is produced by sulfate reduction in/above the SMTZ, and is readily converted to pyrite (Berner, 1981). Previous studies suggest that phosphate concentrations are higher than $\sim$100 $\mu$mol/L in the SMTZ when potential vivianite is observed, so that enhanced phosphate flux could create favorable geochemical conditions for vivianite precipitation below the SMTZ (e.g. Egger et al., 2015; März et al., 2008, 2018). Dissolved Mg concentrations at a nearby site range from 48.4 to 37.7 mmol/L, while dissolved Ca concentrations decrease linearly from the sediment surface to the SMTZ (12.5–2.5 mmol/L) where most of the dissolved Ca is consumed (Ye et al., 2016; Site B). Empirically, Ca tends to be precipitated as CFA independently, while Mg tends to be co-precipitated with vivianite rather than forming Mg phosphate (see R2.1).

Vivianite is apparently lacking in deep-time records. Whether this is an artifact of inefficient surveys and/or non-diagnostic extraction protocols remains to be shown. Generally, authigenic Ca-P concentrations increase with burial depth, likely due to the transfer of P from more labile forms into authigenic apatite, so called sink switching. Sink switching is considered to be the dominant process of apatite authigenesis in marine

sediments worldwide (e.g. Ruttenberg and Berner, 1993; Slomp et al., 1996). Therefore, it is possible that vivianite formed at Site 973-4 is partly converted to authigenic apatite at depth. However, as we stressed in R1.8, the relationship between apatite and vivianite abundances is not straightforward and the details of vivianite preservation and potential transformation remain fundamental unanswered questions.

Finally, there is some element of misunderstanding concerning the origin of the questioned data. Figure 6 displays XRD data derived from handpicked Fe-rich silicates (the grey to green mineral aggregates), whereas the XRD-derived data sourced from handpicked vivianite aggregates is shown in Figure 5a. Vivianite aggregates are only identified below the SMTZ. The vivianite found in the aggregates of Fe-rich silicates at 747 cm depth is intriguing, however, it is not surprising that XRD reveals mineral phases that evade detection optically. As discussed in R1.6, we argue that SMTZ was once shallower and speculate that the vivianite observed at 747 cm was precipitated beneath the SMTZ when it was closer to the sediment water interface. The subsequent downward migration of SMTZ would then be expected to have converted most of the vivianite to iron sulfide erasing the shallow vivianite record. Exactly why the vivianite at 747 cm survived sulfidization is uncertain, however, the textual association with Fe-silicates implies that these phases may have armored the vivianite, preventing conversion to sulfide.

R2.3). Comment: Another concern is the recognition of the current and previous SMTZ. Either porewater or sediment has its own validity in revealing the characteristics and mechanisms of seepage. For example, the geochemical data obtained from the solid fraction of sediments and from authigenic carbonates provide time-averaged information on biogeochemical processes on a timescale of years to centuries. Sediment pore waters and seep-dwelling fauna, on the other hand, provide information on much shorter timescales, spanning from days to months. This issue need to be considered and discussed in discussion. One needs to mention the nature of the seeps. It is well known that seeps are heterogeneous both in time and space. I would find interesting that they describe in a few words the inherent nature of seeps in the introduction section and consequently highlight their findings in the discussion section. The shift of former and current SMTZ is exactly caused by the varying of flux of fluids.

Reply: We agree with R2, and reiterate that pore water and solid phase data resolve processes acting on different timescales. We have discussed this in R1.6. If we had reliable pore water sulfate and methane data, this would constrain the depth of current SMTZ, reflecting seep activity on a timescale of days to years. Solid phase imprints, such as those seen at ∼6–9 m depth (Figs. 7–9), reflect sulfate driven AOM in a SMTZ that has been more-or-less static over decadal to millennial time-scales. Given the constant sedimentation rate at Site 973-4, the position of the SMTZ is controlled by flux of methane-rich fluids from depth. The long-term migration of SMTZ has altered the solid phase records and controlled the distribution of vivianite, creating the mineralogical and geochemical distribution profiles we observe today.

Sincerely, on behalf of all authors,

Jiarui Liu

---

## Author Response (AR1)

**China University of Geosciences**

**College of Marine Science and Technology**

**School of Earth Sciences**

Email: jerryqdc@gmail.com, js-wang@cug.edu.cn

**Biogeosciences**

**Date: October 8, 2018**

**Subject: Post-Review Revision of Manuscript bg-2018-340**

Dear Professor Treude,

Thank you for handing our manuscript—Vivianite formation in methane-rich deep-sea sediments from the South China Sea—for consideration for publication in *Biogeosciences*. We received two generally positive reviews from the solicited reviewers whose efforts, we believe, have helped to clarify and improve the quality of the revised manuscript. Accordingly, where possible we have followed the reviewers' recommendations and amended the manuscript.

Since the initial submission, by mutual consent, we have opted to exchange the order of the second and third authors. Hereby we submit the revised manuscript bg-2018-340 with the full consent of the named authors (Jiarui Liu, Gareth Izon, Jiasheng Wang, Gilad Antler, Zhou Wang, Jie Zhao and Matthias Egger). All the authors declare that the work is their own, it is not under consideration elsewhere, and it is not compromised via any confliction.

With this letter, we append the initial comments from each reviewer (orange), followed by our response (black). We include an explanation of how and where each point raised by the reviewers was incorporated into the manuscript alongside a highlighted manuscript to explicitly locate our changes.

Yours sincerely,

Jiarui Liu and Jiasheng Wang

on behalf of all named co-authors

**Response to the Reviewers: Manuscript bg-2018-340**

**Reviewer 1:** We thank Dr. März for his comments and insight. His requests have helped to clarify the text and highlight omissions from our initial submission. We reply to each of his comments in turn and revise the manuscript accordingly.

R1.1). Comment (sampling of sediments): It is unclear from the text (Methods) how the core sections were processed after sectioning. They were transferred into cool storage. What happened next? Were the sections split on board and samples taken immediately? Were samples stored frozen and/or anoxic to avoid pyrite oxidation prior to freeze-drying? This is crucial for the determination of reactive Fe fractions. Sampling of pore waters: Were pore waters extracted using rhizons? Were they extracted from the closed core by drilling holes into the liner, or were they extracted from the split core surfaces? How long before pore water sampling were the sections split? This is crucial not only for methane and sulphate concentrations, but for any volatile and/or redox sensitive species, i.e., $Fe^{2+}$, $HS^-$, $HPO_4^{2-}$. For example, pore water $Fe^{2+}$ could precipitate as Fe (oxyhydr)oxides and adsorb pore water $HPO_4^{2-}$.

Reply: We acknowledge this initial oversight and agree that more detail should be included to clarify how the sediments/pore waters were handled and collected.

With respect to core handling, after sectioning on board, the cores were stored below 4 °C. The sections were then split and subsampled immediately after the cruise. Sub-samples were stored frozen (−20 °C) prior to freeze-drying to minimize pyrite oxidation. The obvious peak of Fe-oxides at ~900 cm depth (Fig. 8a) may include part of acid volatile sulfur (AVS) pool since AVS oxidizes to dithionite- and oxalate-extractable Fe phases during freeze-drying or exposure to air. It is difficult to quantify the influence of pyrite oxidation on the reactive Fe distributions at Site 973-4 but we anticipate this to be minimized via our sample handling. The clarifying text can be found in the amended manuscript on lines 16–19 on page 4. Acknowledgement of potential post-recovery oxidation can now be found on lines 7–9 on page 8. This was included in the initial submission.

Unfortunately, the pore waters from Site 973-4 were not immediately extracted; instead, they were extracted several months later via centrifugation of previously frozen sub-samples. Consequently, the pore water chemistry of Site 973-4 (Fig. 2b) is more likely to reflect post-recovery oxidation or contamination rather than sediment-housed processes. For example, volatile sulfide loss and/or sulfide oxidation could have modified the distribution of sulfur species. Alternatively, as R1 suggests, precipitation of dissolved Fe could influence the distributions of Fe-sensitive species. It is for this reason that we opted to deemphasize the importance of the published pore water chemistry in favor of data

derived from proximal sites (Fig. 2b; see R1.2). A clarifying paragraph beginning on line 23 (page 5) has been added detailing how the pore water data from the surrounding sites was generated.

R1.2). Comment (pore water data and SMTZ definition): In general, I am not sure if I trust the practice of inferring pore water geochemistry from neighboring sites, especially since no further information is provided about these. How far away are these nearby sites, and were they affected by the same paleo-depositional processes as the study site? Are there distinct similarities in lithology, sedimentation rates etc. that would warrant the "import" of pore water data from these sites? Some of this information can be extracted/inferred from Figure 2 but should be explained in the text as well. From each of the nearby sites, there are only 1–2 methane data points available just below the SMTZ. It should be highlighted in the text that (whether?) this is sufficient to define the SMTZ position. I would also shift the upper boundary of the SMTZ upwards, to the depth where the first methane-free sample was encountered. Also, the methods of pore water sampling and data generation at these nearby sites should be explained.

Reply: We agree with R1's concerns but, given the complications concerning the quality of the pore water data at Site 973-4 (See R1.1), we believe that synthesizing pore water data from the surrounding sites is likely to be more representative. We stress that we do not advocate this approach to replace pore water analysis but, in its absence, we argue that pore water data from nearby sites when combined with solid-phase distributions from Site 973-4 allow us to estimate an approximate position of the SMTZ with sufficient accuracy for the purposes of our manuscript.

Sites from which we obtained pore water data are separated from Site 973-4 by a few kilometers. These sites are lithologically similar and dominated by silty clay. Unfortunately, age constraints and sedimentation rates are lacking at the other sites. The pore water chemistry from the combined sites, however, is analogous, indicating that each site was likely affected by the same depositional process(es). Solid phase distributions at Site 973-4 support this inference. At the sulfidization front, sulfide is known to diffuse out of the SMTZ promoting reductive dissolution of Fe-oxides and the sequestration of elemental sulfur and Fe monosulfides ($3H_2S + 2FeOOH \rightarrow S^0 + 2FeS + 4H_2O$). At Site 973-4 there are two pronounced peaks of elemental sulfur at 730 and 880 cm depth (Liu et al., In prep), while AVS concentrations reach their maximum at ~900 cm depth (Fig. 9b). Accordingly, we combine these site-specific solid-phase records, with the "imported" pore water data to infer the SMTZ is currently around 700–880 cm depth at Site 973-4. As R1 suggested, however, the first methane-free samples appear at 640 cm depth. Considering the first elemental sulfur peak at 730 cm depth, we stress that it is more appropriate to define the upper boundary of the SMTZ as a rough depth for the purposes of our manuscript, i.e. around 700 cm depth.

Given following discussion, and the need to draw on the data of others, we agree with R1 that the pertinent details of pore water sampling and analysis should also be included in the manuscript. Clarifying text to rectify this oversight begins on line 23, page 5. The new text reads: Pore water extraction from Site DH-CL11 was conducted on shore via

centrifugation, exploiting sample aliquots that had been immediately taken and stored under vacuum at −80 °C (Lin et al., 2017). Pore water samples from Site B and Site HD319 were collected immediately after recovery by vacuum extraction (Lu et al., 2012; Ye et al., 2016). At each site, the pore water sulfate contents were determined by ion chromatography; whereas, pore water methane concentrations were determined via sediment plug sampling and gas chromatographic analysis of the resultant headspace gas (Lin et al., 2017; Lu et al., 2012; Ye et al., 2016).

R1.3). Comment (iron extraction): Please cite a reference that defines Fe phases extracted by an anoxic 0.5M HCl solution. I am not actually sure this extraction method has been well calibrated using different Fe minerals.

Reply: The anoxic 0.5 M HCl extraction was adopted from Holmkvist et al. 2011 and 2014 (GCA). More than 97% of amorphous iron (hydro)oxides and ferrihydrite can be extracted, while other Fe-oxides minerals are apparently recalcitrant to this extractant (Wallmann et al., 1993, Limnology and Oceanography; Haese et al., 1997, GCA). Compared to the more well-known dithionite method (e.g. Poulton and Canfield, 2005), the HCl method has not been well calibrated. We added the omitted references on line 31–32 of page 4. An additional sentence has also been added on line 9–11 page 5.

R1.4). Comment (appropriate description of methods): Lots of data are shown (AVS, pyrite, S isotopes, Fe-bound P, Ca-P etc.) that are not covered in the Methods section at all. Even if they were published before, a brief account of how these data were generated is required in this manuscript. Were the analyses conducted on splits of exactly the same samples, or nearby samples, or were samples taken at different times? I would also defer from making speculations about what certain data would look like if they had been generated (e.g., pore water $HPO_4^{2-}$).

Reply: Following R1's advice, a brief description of how the various published datasets were generated has been included in the amended manuscript. The new paragraph is found on line 31, page 5, which reads: Previously published data was determined using separate aliquots of the same subsamples exploited herein. This earlier work followed established protocols and full methodological details are provided in the respective papers. Accordingly, only a brief description is provided here. Acid volatile sulfide was liberated via HCl distillation and trapped by zinc acetate, its concentration was then determined spectrophotometrically (Zhang et al., 2014). Pyrite aggregates were handpicked from the coarse-fraction (> 65 µm) with its abundance expressed relative to the coarse grains. The sulfur isotopic composition of handpicked pyrite was determined directly via flash combustion using a Delta V Plus isotope ratio mass spectrometer (IRMS; Lin et al., 2015). The solid-phase distribution of P was revealed through the SEDEX sequential extraction scheme (Ruttenberg, 1992). Fe-bound P and authigenic carbonate fluorapatite were extracted by citrate-bicarbonate-dithionite and Na-acetate buffer, respectively (Zhang et al., 2018b). The carbon isotopic composition of total inorganic carbon was determined after treatment with phosphoric acid using a Finnigan MAT-252 IRMS (Ou, 2013; Zhang et al., 2018c). Finally, magnetic susceptibility data was generated using a MFK1-FA kappameter (Lin et al., 2017a).

All analyses were conducted on aliquots of the same sample powder, ensuring the data are comparable. As detailed on line 17–19, page 4, subsamples of the core were split into two and the second aliquot was homogenized for chemical analysis. Although the analyses were made at different times, the powdered samples were kept frozen to minimize oxidation.

Following R1's advice, speculative aspects have been removed from the amended manuscript.

R1.5). Comment (the Ca-P fraction): A lot of Ca-P is found below the SMTZ, more than Fe-bound P. This is not discussed sufficiently. Is part of the $HPO_4^{2-}$ that is (tentatively) liberated in the SMTZ precipitated as authigenic apatite? Is this supported by pore water Ca and F profiles (if not at this site, maybe at the nearby sites)? And could the formation of Ca-P be related to the sulphidisation front that consumed part of the Fe-AOM-derived $Fe^{2+}$ and precipitated it as AVS? In other words, does the activity of a sulphidisation front put a constraint on how much Fe(II) phosphate can precipitate below the SMTZ? Are there any estimates of the kinetics of these potentially competing authigenic processes of $Fe^{2+}$ removal from the sub-SMTZ pore waters? This would be very interesting, as it would allow to better link the rate of $HS^-$ production in the SMTZ (which is largely controlled by sulphate diffusion from above and methane delivery from below), the amount of reactive Fe oxides beneath the SMTZ, and the potential to form different authigenic P phases beneath the SMTZ.

Reply: This is an interesting point, and one that warrants inclusion in the discussion of the amended manuscript, albeit still speculative due to the lack of appropriate data. We have added an extra section to the manuscript (*Long-term vivianite preservation—An outstanding question*) on page 14 which includes the following points:

As pointed out by R1, authigenic Ca-P is the major P sink at Site 973-4 (~55%), in agreement with its globally estimated importance. Indeed, Ca-P concentrations are much higher than Fe-bound P throughout the studied core. It is well known that a steady supply of phosphate, fueled by the reductive dissolution of Fe-oxides, fosters thermodynamically favorable conditions capable of promoting apatite authigenesis in proximity to the SMTZ (Egger et al., 2015a, Fig. S8; März et al., 2018). Thus, as suggested by R1, it is possible that some of the SMTZ-sourced phosphate is consumed and precipitated as authigenic apatite, providing the other necessary chemical constituents are available. While there are no reliable pore water data from Site 973-4 (R1.1), dissolved $Ca^{2+}$ concentrations from Site B are consistent with ongoing apatite authigenesis, with a linear down-core decrease from the sediment-water interface to the upper part of the SMTZ (12.5–2.5 mM) where most of the $Ca^{2+}$ has been consumed (Ye et al., 2016). To our knowledge, however, there are no available dissolved $F^-$ and $PO_4$ data to unequivocally confirm apatite precipitation rather than other calcium-harboring phases.

Vivianite formation below the SMTZ depends on $PO_4$ and $Fe^{2+}$ availability, which we suggest are mainly supplied from $PO_4$ liberation in the SMTZ and the upward flux of $Fe^{2+}$ from iron reduction at depth. Kinetics notwithstanding, providing

there is sufficient $Ca^{2+}$ and $F^-$, apatite will remove $PO_4$ competing and effectively limiting vivianite precipitation. At Site 973-4 vivianite occurs below ~920 cm depth suggesting that even if apatite authigenesis is a competing process, $Ca^{2+}$ and $F^-$ have likely been consumed at this depth. Importantly, vivianite authigenesis occurs below the sulfidization front (~900 cm) implying that FeS precipitation is kinetically favored relative to vivianite precipitation immediately below the SMTZ. We speculate, therefore, that the pore water chemistry and the activity of sulfidization front will influence the depth of vivianite formation. Intensified activity at the sulfidization front would liberate more phosphate, however, the extra sulfide flux may serve to nullify the ascent of $Fe^{2+}$ from depth, especially as highly reactive iron is consumed, preventing vivianite formation, or forcing it deeper into the sediment-pile.

Unfortunately, we do not have the data to definitively test the hypothesized competing role of apatite and FeS authigenesis in vivianite precipitation. However, any process that serves to deplete pore water $PO_4$ and $Fe^{2+}$ will hypothetically compete with vivianite formation. Future work in/around the study area should focus on generating comprehensive pore water and solid phase data sets, which when coupled with reactive transport models, will better link the rate of $HS^-$ production in the SMTZ, the amount of reactive Fe-oxides beneath the SMTZ, and the potential to form different authigenic P phases.

R1.6). Comment ("deepening" of the SMTZ): The suggestion of a previously shallower SMTZ in section 5.1 comes out of the blue, without any specific reasoning for why the authors think this was the case. I do not necessarily disagree with this hypothesis, but the authors need to give some supporting arguments, e.g., maybe the fact that pyrite exists above the SMTZ (although this could also be formed by organoclastic sulphate reduction). They then need to better develop how such a deepening of the SMTZ would have affected all the described geochemical parameters, and if observations agree with expectations. Finally, they need to relate the migration to the SMTZ to depositional/environmental processes – if it was not caused by changes in sedimentation rate, was it changes in methane flux from below?

Reply: If we accept that the current SMTZ is around 700–880 cm depth based on the arguments presented in R1.2, we can speculate on the evolution of the SMTZ at Site 973-4. The preservation of Fe-oxides and absence of pyrite below the SMTZ were most likely caused by the rapid upward migration of the SMTZ from below 14m (i.e. deeper than the core) to middle of our record. The resultant short exposure of Fe-oxide minerals to sulfidic conditions led to incomplete reduction and sulfidization of the reactive iron pool (März et al., 2018). After its ascent, the SMTZ apparently stabilized around 560–880 cm, leaving pronounced imprints in the pyrite, Fe-oxide, Fe-silicate, total inorganic carbon and magnetic susceptibility profiles (Figs. 7–9). It should be noted that pyrite formed by organoclastic sulfate reduction usually features more negative $\delta^{34}S$ values (−50‰) such as those seen at ~300–560 cm depth. Whereas, the $\delta^{34}S$ excursion seen at ~560–880 cm depth is best explained by sulfate driven AOM at and above the current SMTZ. Assimilating these observations,

we suggest that the SMTZ might have slowly migrated downward from ~560–700 cm to its current position at ~700–880 cm depth.

This migration pattern may explain why we observed vivianite at 747 cm depth (Fig. 6). The paleo-SMTZ may have been shallower (e.g. ~560–700 cm depth) allowing vivianite to precipitate at shallower depths (e.g. 747 cm depth). We speculate that most of the vivianite formed at/around the current SMTZ was readily converted to Fe-sulfide phases via reaction with $H_2S$ during the subsequent downward migration of the SMTZ, concentrating vivianite below the current SMTZ. Consequently, if the SMTZ continued to descend within the sediment pile, we would expect that the pronounced solid-phase imprints observed at ~560–880 cm would also descend to sediments below the current SMTZ.

Given the largely invariant sedimentation rate at Site 973-4 over the duration of the core (Fig. 2a), we hypothesize that the migration history of the SMTZ was driven by changes in the methane flux. During the sea-level low stand during the Last Glacial Maximum, the resulting low hydrostatic pressure would have destabilized any underlying methane clathrates, enhancing methane fluxes while promoting a rapid upward migration of the SMTZ (Borowski et al., 1996). The subsequent Holocene sea-level rise, however, would have had the opposite effect, diminishing methane fluxes and instigating the slow downward migration of the SMTZ to its current position. Our observations are consistent with those of others who suggest that the SMTZ needs to be static (i.e. ~560–880 cm depth) for thousands of years to allow the development of significant authigenic Fe(II) phosphate mineral enrichments below the SMTZ (März et al., 2018).

The manuscript has been substantially modified to discuss the evolution of the SMTZ at Site 973-4. These amendments are mainly in Section 5.1 (page 8). Specifically, among the addition of a few additional sentences, we have added an extra paragraph beginning on line 21 page 9 to expand on the evidence for SMTZ migration and our favored driving mechanism. Toward the end of the following paragraph we have introduced the consequences of SMTZ migration on the solid-phase geochemical records.

R1.7). Comment (missing reference): Please cite März et al. (2008, 2018; Marine Geology) when stating that a lot of the vivianite is likely finely disseminated. In general, wherever the authors speculate about the wide-spread occurrence of vivianite in methane-rich continental margin sediments, they need to cite the new study by März et al. (2018) (Marine Geology) that comes to exactly the same conclusion.

Reply: We apologize for the omission of the reviewer's most recent work. We had submitted our paper before the 2018 study had been formally published. As suggested, in our amended manuscript, we have cited März et al. (2008 and 2018) to fully credit the importance of finely disseminated vivianite in methane-rich continental margin settings.

R1.8). Comment (vivianite to Ca-P sink switching): When looking at the Ca-P profile, one cannot help but notice that there is an increase to higher values from the SMTZ to the bottom of the core, while Fe-P and vivianite seem to decrease

in parallel. This raises the question of the long-term stability of vivianite in the sedimentary record, and its use as paleo SMTZ marker. In fact, vivianite is hardly ever found in older sediments/sedimentary rocks where carbonate fluorapatite is by far the dominant mineral, so there is a strong argument that vivianite is (at least partly) transformed into something else, maybe authigenic apatite?

Reply: The question concerning vivianite's longevity in the sedimentary record is a good one and, indeed, it is an outstanding question that needs to be addressed more generally. Unfortunately, in the context of the available data, we cannot unequivocally answer the posed questions and thus we opted to expand the discussion to encompass the available observations and interpretations. However, given the ambiguity, we also elected to "tone-down" our language to convey uncertainty in the amended manuscript. Again, additional text can be found in the new section included on page 14.

The increase in Ca-P concentrations noted by R1 may reflect the conversion of labile P-species into authigenic apatite—so-called sink switching. Sink switching is considered to be the dominant global process of apatite authigenesis in marine sediments (e.g. Ruttenberg and Berner, 1993; Slomp et al., 1996). Whether vivianite is converted to authigenic apatite at depth, while possible, remains an open question. Moreover, if it were a simple conversion then one would predict a linear increase in apatite with a concomitant decrease in vivianite. Closer inspection of the relevant solid phase records reveals that this is not the case and, in fact, the relationship between apatite and vivianite concentrations is ambiguous, and largely decoupled, at Site 973-4.

Alternatively, the "increase" in authigenic Ca-P from the SMTZ to the bottom of the core may reflect non-vivianite-related apatite authigenesis at depth. Upward fluxes of $Ca^{2+}$ and $F^-$ to the SMTZ have been observed in continental margin settings (e.g. Clemens et al., 2016; März et al., 2018), which would provide the necessary chemical constituents to promote sub-SMTZ apatite authigenesis. As discussed in R1.5, if apatite is a more favorable phosphate sink, then precipitation of this mineral phase could hypothetically curtail vivianite authigenesis. In this scenario, although the two phosphate phases are linked, they are linked through $Ca^{2+}$ (and $F^-$) availability rather than stability and conversion of one phase to another.

The apparent decrease in vivianite with depth could also be completely decoupled from apatite authigenesis. For example, vivianite is sensitive to oxic conditions, thus it is possible that vivianite in deep-time records has been oxidized to Fe-oxides (e.g., hematite; Berner, 1981), Fe(III) phosphate (e.g., koninckite; März et al., 2008a) or some other unknown phase(s). Pseudomorphs of vivianite are apparently common in the deep-time record, providing at least some evidence for the eventual transformation and stabilization of vivianite (e.g. metavivianite; Rodgers, 1986). Equally, the perceived absence of vivianite in aged sedimentary archives could be an artifact of insufficient surveys. Our work, as well as that of R1 (März et al., 2018), has shown that vivianite is likely a finely disseminated phase and thus may have escaped

detection in many studies. Moreover, typically employed extraction schemes co-extract vivianite, meaning that Fe-P in deep time records could have been wrongly ascribed, and could be vivianite (c.f., März et al., 2008b).

Future work, targeting long cores coupling aqueous- and solid-phase analysis, is required to definitively test our hypotheses linking vivianite authigenesis and Fe-AOM activity, as well as to examine the long-term stability of vivianite in the sedimentary record. More routine application of XRD and modification of solid-phase extraction protocols is also necessary to answer questions relating to the geological significance of vivianite.

R1.9). Comment (importance of findings to the geological past): The link between vivianite formation below the SMTZ in Fe-AOM affected sediments and the potential importance of these processes in the pre-GOE oceans is not well-developed. The actual effects of Fe(II) phosphate formation in an Fe-rich ocean are not discussed at all, only the potential importance of Fe-AOM is mentioned (which is not wrong, but also does not reflect the main story of the research presented here). It would be much more interesting to develop what impact Fe(II) phosphate formation under ferruginous conditions might have had on the marine P cycle. These considerations do not only apply to the pre GOE ocean; as first proposed by März et al. (2008) (GCA) and further developed by Poulton and Canfield (2011) (Elements), ferruginous conditions existed periodically in the Mesozoic (and probably throughout the Phanerozoic) as well. And as März et al. (2008) (GCA) pointed out, the precipitation of Fe(II) phosphates occurred under ferruginous conditions during the deposition of Cretaceous black shales on Demerara Rise, sequestering P from the water column and putting a constraint on the anoxia productivity feedback loop. While we still do not understand enough about the details, and potential effects, of Fe(II) phosphate formation under ferruginous conditions, these earlier studies should be referenced appropriately.

Reply: These are interesting and valid points. We have added clarifying text from line 8 (page 15) to line 2 on the following page. As detailed by R1, the seawater chemistry was vastly different between ancient oceans and their modern counterparts. In our original manuscript, we specifically referred to the predominantly ferruginous state of pre-GOE oceans, yet we concede that ferruginous conditions were also a periodic feature post-GOE oceans as well (März et al., 2008b; Poulton and Canfield, 2011). The latter scenario is subtly different from the former, however, due to ingrowth of the seawater sulfate reservoir (Canfield et al., 2000). Accordingly, we have extended the discussion, fortifying the stance of others while more completely developing our hypotheses concerning the importance of vivianite, and its wider role in P-cycling in ancient oceans.

Where basal waters are oxygenated, we expect a similar pore water pattern to that observed at Site 973-4 or other modern open marine settings (Fig. 10), where vivianite is dominantly formed below the SMTZ. If oxygen concentrations drop, however, vivianite authigenesis is expected to be different with implications for P-cycling and productivity. These intricacies, however, remain to be fully explored.

Under ferruginous (anoxic and $Fe^{2+}$-rich) conditions dissolved phosphate would be adsorbed to and/or co-precipitated with Fe-oxides (März et al., 2008b), or even precipitated as vivianite directly if $Fe^{2+}$ and $PO_4$ concentrations were sufficiently high (Dijkstra et al., 2018b). This P shuttle is (at least) partially eradicated if euxinic conditions develop in the water column caused by sulfidization of Fe-oxides and/or vivianite presumably governed by particle settling kinetics. After settling, the fate of the P-rich Fe-oxides and vivianite particles would be dictated by the prevailing conditions in the diagenetic environment. Under sulfate limited conditions P removed from the water column is likely to be retained in the sediment-pile, throttling productivity via a negative feedback (März et al., 2008b). When sulfate is available, however, reductive dissolution of Fe-oxides will liberate phosphate to the pore water which may be lost or retained dependent on the locus of dissolution and the availability of other important dissolved species ($Ca^{2+}$, $Fe^{2+}$, $Mg^{2+}$, $F^-$) and reductants (e.g., organic matter, $CH_4$).

Applying these principles to our understanding of Earth History is somewhat speculative but nonetheless interesting, with broad implications for P-cycling. In the largely low-sulfate ferruginous oceans prior to the GOE, the development of euxinia would have been scarce, enhancing phosphate shuttling to the sediment. If this phosphate was efficiently retained on a global-scale, then phosphate availability in these ancient oceans would have been low, especially considering the reduction in oxidative weathering (Reinhard et al., 2017). By contrast, throughout Earth's history euxinic marine environments have become more prevalent—a consequence of rising sulfate concentrations (Poulton, 2017). As discussed, euxinic conditions promote P recycling and its return to the water column; whereas P is more recalcitrant in ferruginous environments. In a simple sense, therefore, euxinic conditions are touted to be quasi self-sustaining (via a productivity feedback) whereas ferruginous conditions are not. In a wider Earth System sense, however, these gross generalizations may not be valid, and warrant further investigation. Sustained euxinia, for example, could lower sulfate inventories favoring the development of ferruginous conditions if $S/Fe_{HR}$ ratios drop below 1.8 (e.g., Poulton and Canfield., 2011). Furthermore, unless reactive iron enrichments are solely derived from hydrothermal emanations, simple mass balance constraints dictate that sedimentary iron enrichments are unlikely to occur throughout an anoxic ocean and the source region must be depleted in reactive iron (e.g., Poulton and Canfield., 2011). Consequently, to understand the global effects of oxygen deficiency on P-cycling we must more completely understand the local controls on P-cycling and how oxygen deficiency developed on an event-by-event basis to successfully integrate these processes in an Earth system model.

**Reviewer 2:** We would like to express our gratitude to R2. Most of the points raised here were also raised by R1, we have attempted to cross reference between our responses to prevent repetition. We reply to each of the reviewer's comments in turn and have amended the manuscript as necessary.

R2.1). Comment: Although the chemical composition of the pore-water (pH, $PO_4^{3-}$, $Ca^{2+}$ etc.) is lacking, a simple geochemical calculation on what kind of minerals (FeS, CFA, $Mg_3(PO_4)_2$, $Fe_3(PO_4)_2$) are expected to be precipitated in/above/under SMTZ would make the P-cycle in methane-rich environments clearer.

Reply: Pore water chemistry is essential to conduct a comprehensive series of calculations on mineral precipitation in marine sediments. As discussed in R1.1, we lack robust pore water data and thus we draw on PHREEQC calculations of pore water saturation indexes from other sites (Egger et al., 2015a, Baltic Sea, Fig. S8; März et al., 2018, open marine margins) to address this comment.

Hydroxyapatite saturation indexes above 0 suggest that the steady supply of phosphate from reductive dissolution of Fe-oxides creates thermodynamically favorable conditions for carbonate fluorapatite (CFA) precipitation around the SMTZ, whereas CFA formation appears to be a minor sink for pore water phosphate below the SMTZ (März et al., 2018). Higher saturation indexes for vivianite are typically obtained below the SMTZ. Here, the upward flux of $Fe^{2+}$ from iron reduction and downward flux of phosphate from the reductive dissolution of Fe-oxides create favorable geochemical conditions for vivianite precipitation below the SMTZ, consistent with the vivianite distribution observed at Site 973-4. The solubility product constant $K_{sp}$ of Mg phosphate is $10^{-24}$, while the $K_{sp}$ of vivianite is $10^{-36}$ (Nriagu, 1972), making the former mineral more soluble. Moreover, Mg concentrations are only about three orders of magnitude higher than those of Fe in marine sediments (e.g. Hu et al., 2015, Site D-5). Consequently, Mg phosphate precipitation is unlikely. Rather, $Mg^{2+}$ is more likely to be co-precipitated with vivianite, forming Mg-rich vivianite below the SMTZ. This is clarified on line 18–21, page 11 and the following paragraph.

Besides phosphate minerals, Fe sulfides are enriched in and above the SMTZ. Pyrite is a common mineral in marine sediments where dissolved sulfide, produced by either organoclastic sulfate reduction (above the SMTZ) or sulfate driven anaerobic oxidation of methane (within the SMTZ), is trapped by Fe-oxides, forming Fe monosulfide. Fe monosulfides are gradually converted to pyrite as a permanent sink for sulfur. The acid volatile sulfur peak at ~900 cm depth contains large amounts of Fe monosulfides, and is referred to as the sulfidization front (S-front). The dissolved Fe is sourced either from the reductive dissolution of Fe-oxides by sulfide at the same depth or iron reduction at depth. Consequently, the formation of Fe sulfide is modulated by sulfate penetration depth. Clarifying text is found on line 17–19, page 8.

R2.2). Comment: Why is vivianite only observed/common under SMTZ (Fig 7a) instead of forming in/above SMTZ. What concentration of $HPO_4^{2-}$ is expected when the Mg-rich vivianite is observed (Fig. 4) (The $K_{sp}$ of $Mg_3(PO_4)_2$ are

three orders of magnitude higher than $Ca_3(PO_4)_2$, but the $[Mg^{2+}]$ in pore-water are one order of magnitude higher than $[Ca^{2+}]$). Vivianite is lacking in ancient record. Would the vivianite formed here be converted to $Ca_3(PO_4)_2$ or what conditions are necessary for vivianite preservation in the sedimentary record to serve as a proxy for methane rich environment or Fe-AOM activity? In Fig 6, vivianite is only observed at 747 cm? Why is there no vivianite below the 900 cm depth according to the XRF result? There is an inconsistency between vivianite content determined via XRF and handpicking.

Reply: We did not identify any macro-vivianite within or above the SMTZ. Generally, vivianite is unstable in the presence of sulfide that is produced by sulfate reduction in/above the SMTZ, and is readily converted to pyrite (Berner, 1981). Previous studies suggest that phosphate concentrations are higher than ~100 µM in the SMTZ when potential vivianite is observed, so that enhanced phosphate flux could create favorable geochemical conditions for vivianite precipitation below the SMTZ (e.g. Egger et al., 2015; März et al., 2008a, 2018). Dissolved Mg concentrations at a nearby site range from 48.4 to 37.7 mM, while dissolved Ca concentrations decrease linearly from the sediment surface to the SMTZ (12.5–2.5 mM) where most of the dissolved Ca is consumed (Ye et al., 2016; Site B). Empirically, Ca tends to be precipitated as CFA independently, while Mg tends to be co-precipitated with vivianite rather than forming Mg phosphate (see R2.1).

Vivianite is apparently lacking in deep-time records. Whether this is an artifact of inefficient surveys and/or non-diagnostic extraction protocols remains to be shown (see text in Section 5.4, page 14). Generally, authigenic Ca-P concentrations increase with burial depth, likely due to the transfer of P from more labile forms into authigenic apatite. This sink switching is considered to be the dominant process of apatite authigenesis in marine sediments worldwide (e.g. Ruttenberg and Berner, 1993; Slomp et al., 1996). Therefore, it is possible that vivianite formed at Site 973-4 is partly converted to authigenic apatite at depth. However, as we stressed in R1.8, the relationship between apatite and vivianite abundances is not straightforward and the details of vivianite preservation and potential transformation remain fundamental unanswered questions.

Finally, there is some element of misunderstanding concerning the origin of the questioned data. Figure 6 displays XRD data derived from handpicked Fe-rich silicates (the grey to green mineral aggregates), whereas the XRD-derived data sourced from handpicked vivianite aggregates is shown in Figure 5a. Vivianite aggregates are only identified below the SMTZ. The vivianite found in the aggregates of Fe-rich silicates at 747 cm depth is intriguing, however, it is not surprising that XRD reveals mineral phases that evade detection optically. As discussed in R1.6 (see from line 5–12 on page 10), we argue that SMTZ was once shallower and speculate that the vivianite observed at 747 cm was precipitated beneath the SMTZ when it was closer to the sediment water interface. The subsequent downward migration of SMTZ would then be expected to have converted most of the vivianite to iron sulfide erasing the shallow vivianite record.

Exactly why the vivianite at 747 cm survived sulfidization is uncertain, however, the textual association with Fe-silicates implies that these phases may have armored the vivianite, preventing conversion to sulfide.

R2.3). Comment: Another concern is the recognition of the current and previous SMTZs. Either porewater or sediment has its own validity in revealing the characteristics and mechanisms of seepage. For example, the geochemical data obtained from the solid fraction of sediments and from authigenic carbonates provide time-averaged information on biogeochemical processes on a timescale of years to centuries. Sediment pore waters and seep-dwelling fauna, on the other hand, provide information on much shorter timescales, spanning from days to months. This issue needs to be considered and discussed in discussion. One needs to mention the nature of the seeps. It is well known that seeps are heterogeneous both in time and space. I would find interesting that they describe in a few words the inherent nature of seeps in the introduction section and consequently highlight their findings in the discussion section. The shift of former and current SMTZ is exactly caused by the varying of flux of fluids.

Reply: We agree with R2, and reiterate that pore water and solid phase data resolve processes acting on different timescales (line 24–30, page 8). We have discussed this in R1.6. If we had reliable pore water sulfate and methane data, this would constrain the depth of current SMTZ, reflecting seep activity on a timescale of days to years. Solid phase imprints, such as those seen at ~6–9 m depth (Figs. 7–9), reflect sulfate driven AOM in a SMTZ that has been more-or-less static over decadal to millennial time-scales. Given the constant sedimentation rate at Site 973-4, the position of the SMTZ is controlled by flux of methane-rich fluids from depth. The long-term migration of SMTZ has altered the solid phase records and controlled the distribution of vivianite, creating 
[revised manuscript text omitted]